# Earth transformed: detailed mapping of global human modification from 1990 to

# <sub>3</sub> 2017

4 David M. Theobald<sup>1,2</sup>, Christina Kennedy<sup>3</sup>, Bin Chen<sup>4</sup>, James Oakleaf<sup>3</sup>, Sharon Baruch-Mordo<sup>3</sup>, Joe

5 Kiesecker<sup>3</sup>

- <sup>6</sup> <sup>1</sup>Conservation Planning Technologies, Fort Collins, CO 80521, USA
- <sup>2</sup>Department of Fish, Wildlife, and Conservation Biology, Colorado State University, Fort Collins, CO
- 80523, USA
- <sup>3</sup>Global Lands Program, The Nature Conservancy, Fort Collins, CO 80524, USA
- <sup>4</sup>Department of Land, Air and Water Resources, University of California, Davis, CA 95616, USA

## 11

Correspondence to: David M. Theobald, Ph.D. (<u>dmt@davidmtheobald.com</u>)

## 13

# 16 Abstract

- Data on the extent, patterns, and trends of human land use are critically important to support global
- and national priorities for conservation and sustainable development. To inform these issues, we
- created a series of detailed global datasets for 1990, 2000, 2010, and 2015 to evaluate temporal and
- spatial trends of land use modification of terrestrial lands (excluding Antarctica). We found that the
- expansion and increase of human modification between 1990 and 2015 resulted in 1.6 M km<sup>2</sup> of
- natural land lost. The percent change between 1990 and 2015 was 15.2% or 0.6% annually -- about 178
- $23~\mbox{km}^2$  daily, or over 12 hectares each minute. Worrisomely, we found that the global rate of loss has
- increased over the past 25 years. The greatest loss of natural lands from 1990-2015 occurred in
- Oceania, Asia, and Europe, and the biomes with the greatest loss were mangroves, tropical &
- subtropical moist broadleaf forests, and tropical & subtropical dry broadleaf forests. We also created
- a contemporary (~2017) estimate of human modification that included additional stressors and found
- that globally 14.6% or 18.5 M km<sup>2</sup> ( $\pm$ 0.0013) of lands have been modified -- an area greater than Russia.
- Our novel datasets are detailed (0.09 km<sup>2</sup> resolution), temporal (1990-2015), recent (~2017),
- comprehensive (11 change stressors, 14 current), robust (using an established framework and
- incorporating classification errors and parameter uncertainty), and strongly validated. We believe
- these datasets support an improved understanding of the profound transformation wrought by
- human activities and provide foundational data on the amounts, patterns, and rates of landscape
- change to inform planning and decision making for environmental mitigation, protection, and
- restoration.
- The datasets generated from this work are available at <u>https://zenodo.org/record/3963013</u> (Theobald
- et al. 2020).

# 39 1 Introduction

Humans have transformed the earth in profound ways (Marsh 1885; Jordan et al. 1990; Vitousek et al.

- 1997), contributing to global climate change (IPCC 2019), causing global habitat loss and
- fragmentation, and contributing to declines in biodiversity and critical ecosystem services (IPBES
- 2019). Addressing the consequences of rapid habitat loss and land use change are essential for
- implementation of various international initiatives, including the Convention on Biological Diversity
- 2020 Aichi Biodiversity targets, the United Nations 2030 Sustainable Development Goals (esp. Goal
- 15; Secretariat of the Convention on Biological Diversity, 2010), the Bonn Challenge (Verdone & Seidl,
- 2017), and the Global Deal for Nature (Dinerstein et al. 2019). Foundational to addressing these goals
- is a firm understanding of the rates, trends, and amount of these land use changes. Efforts to date
- have focused on historical patterns (Klein Goldewijk et al. 2007, 2017; Ramankutty et al. 2008; Ellis
- 2018) or have been limited due to the unavailability of contemporary, temporally comparable, and

high-resolution (< 1 km<sup>2</sup>) data (Venter et al. 2016; Geldman et al. 2019; Kennedy et al. 2019a).

Here we describe a new dataset that maps the degree of human modification of terrestrial

- ecosystems globally, for recent changes from 1990 to 2015, and for contemporary (circa 2017)
- conditions. We mapped human activities that directly or indirectly alter natural systems, which we
- call anthropogenic drivers of ecological stress or "stressors" (following Salafsky *et al.*, 2008;
- Theobald 2013). Similar to other efforts (Sanderson et al. 2002; Theobald 2010, 2013; Geldmann et al.
- 2014; Venter et al. 2016; Kennedy et al. 2019a), we augmented remotely-sensed data with
- traditionally-mapped cartographic features. This is because remotely sensed imagery has limitations
- for this application -- especially prior to ~2010 -- because it can require human-interpretation to
- classify adequately and can miss development features that are obstructed by vegetation canopy or
- are small or narrow features (e.g., towers, wind turbines, powerlines).
- We mapped the degree of human modification based on an established approach that has been
- applied nationally, internationally, and globally (Theobald 2010, 2013; Gonzalez-Abraham et al. 2015;
- Kennedy et al. 2019a). It uses an existing classification system (Salafsky et al., 2008) to: (a) ensure
- parsimony; (b) distinguish two spatial components (area of use and intensity of use); (c) use a
- physically-based measure that is needed to estimate change (Gardner and Urban 2007); (d)
- incorporate spatial and classification uncertainty; and (e) combine multiple stressors into an overall
- measure that assumes additive relationships among stressors and addresses the correlation among
- variables (Theobald 2010). The resulting quantitative estimate of human modification has values
- ranging from 0 to 1 that support robust landscape assessments (Schultz 2001; Hajkowicz and Collins
- 2007).
- 75~ To understand temporal landscape change, we calculated the degree of human modification --
- denoted by H -- for the years 1990, 2000, 2010, and 2015 using methods and datasets that minimize
- noise and bias. Second, we included additional stressors not incorporated previously, including
- disturbance of natural processes due to reservoirs, effects from air pollution, and human intrusion
- (Theobald 2008). Third, we calculated human stressors using up to two orders of magnitude finer

- resolution data (0.09 vs. 1-86 km<sup>2</sup>) than past efforts (Ellis and Ramankutty 2008; Geldmann et al.
- 2014; Haddad et al. 2015; Venter et al. 2016; Geldmann et al. 2019b; Kennedy et al. 2019a). This higher
- resolution reduces the loss of information of the spatial pattern within a pixel, better identifies rare
- features, facilitates the application of these data for species and ecological processes that often
- occur at a fine-scale, and improves the utility and relevance of these products for policy makers,
- decision makers, and land use managers.
- Calculating H as a real value across the full gradient of landscape change is valuable because it can be
- applied rigorously to a variety of questions (Theobald 2010, 2013), including discerning the
- heterogeneity of human uses that are often lumped within broad classes like "urban"; capturing the
- extent and pattern of the agricultural lands typically occurring beyond urban centers and protected
- areas; and delineating areas of low modification: all of which are useful for conservation
- prioritization and planning efforts (Kennedy et al. 2019a, 2019b). Here, we describe the technical
- methods and briefly report on results on the temporal trends and current spatial patterns of human
- modification across all terrestrial lands, continents, biomes and ecoregions . Because conservation
- organizations often use this type of data to focus their activities on specific regions (e.g., Jantke et
- al. 2019), we provide rankings by biome and ecoregion (Dinerstein et al. 2017) and briefly compare
- our results to other available studies.

# 98 2 Methods

## 99 2.1 Overview

- We calculated the degree of human modification using the Direct Threats Classification v2 (Salafsky
- et al. 2008; cmp-openstandards.org), which defines a stressor as the proximate human activities or
- processes that have caused, are causing, or may cause impacts on biodiversity and ecosystems. Table
- 1 lists the specific stressors and data sources we included in our maps: urban/built-up, crop and
- pasture lands, livestock grazing, oil and gas production, mining and quarrying, power generation
- (renewable and non-renewable), roads, railways, power lines and towers, logging and wood
- harvesting, human intrusion, reservoirs, and air pollution.
- To estimate *temporal change* in H from 1990 to 2015, we followed established criteria (Geldmann et
- al. 2014) and included 11 stressors for which we could obtain global data with fine-grained resolution
- (<1 km<sup>2</sup>), and that provided consistent and comparable repeated measurements, especially in
- regards to the data source, methods used, and appropriate time frame (Table 1). We included current
- major roads and railways as a static layer in the temporal maps because in most cases some form of
- road existed prior to our baseline year of 1990 (except for the relatively rare, though important, new
- highway constructed).
- To estimate the current amount of H circa 2017 year (median=2017, min=2012, max=2019), we included
- three additional stressors, including grazing, oil and gas wells, and powerlines. We note that we did
- not map stressors for invasive species or pathogens and genes, geologic events, or climate change.

This was because suitable temporal global data were not available to capture stressors due to

invasive species or pathogens and genes; the majority of geological events are not directly caused by

humans; and climate change is better modeled as separate process distinct from the effects of direct

- human activities and has a plethora of research on this topic (Geldmann et al. 2014; Titeux et al.
- **2016).**

For each stressor *s* we quantified the degree of human modification as:

$$H_s = F_s * p(C_s) * I_s$$
,

(1)

where  $F_s$  is the proportion of a pixel occupied (i.e. the footprint) by stressor s,  $p(C_s)$  is the probability that a stressor occurs at a location to account for spatial and classification uncertainty, and  $I_s$  is the

129 intensity. Importantly, F and I have a direct physical interpretation (Gardner and Urban 2007), are

130 well-bounded and range from 0-1, and values are a "real" data-type. Consequently, H provides the

- basis for unambiguous interpretation to assess landscape change (Hajkowicz and Collins 2007;
- Riitters et al. 2009). Specific formulas used to map raw stressor data as indicator layers are provided

below. Table 2 details our estimates of intensity values for each stressor (modified from Theobald

2013 and Kennedy et al. 2019a), which is used to differentiate land uses that have varying impacts on

terrestrial systems (e.g., grazing is less intensive than mining). Our intensity values were informed by
 standardized measures of the amount of non-renewable energy required to maintain human

- activities (Brown and Vivas 2005) and found to generally correlate with species responses to land use
- where examined (Kennedy et al. 2019a).

We generated datasets that represent temporal changes between 1990 and 2015 and for current

- (~2017) conditions by combining stressor layers using the fuzzy algebraic sum (Bonham-Carter, 1994;
- Malczewski 1999; Theobald 2013), which is calculated as:

$$H = 1 - \prod_{s=1}^{n} (1 - H_s),$$

(2)

where n is the number of stressors (s) included. Of critical importance, the fuzzy sum formula is an *increasive* function that calculates the cumulative effects of multiple stressors in a way that
minimizes the bias associated with non-independent stressors and assumes that multiple stressors
accumulate (Theobald 2010, 2013; Kennedy et al. 2019a). This differs substantially from simple
additive calculations that are commonly used (Halpern et al. 2008; Halpern and Fujita 2013; Venter et
al. 2016), but assume that stressors are independent and results in a metric that is sensitive to the
number of stressors included in the model (Malczewski 1999).

We mapped human modification of all terrestrial lands (excluding Antarctica) and included lands inundated by reservoirs, but excluded other rivers and lakes. An often overlooked but critical aspect to understand human modification is how water is mapped, especially for the interface between land and coastlines, lakes, reservoirs, and large rivers. We mapped non-reservoir areas dominated by water (i.e., oceans, lakes, reservoirs, and rivers) by processing data on ocean from the European Space Agency's Climate Change Initiative program (ESA CCI; 0.15 km, circa 2000) and surface waters using the Global Surface Water dataset (GSW; 30 m; Pekel et al. 2016). We identified inland water

- bodies (i.e. lakes, reservoirs, rivers, etc.) using ESA CCI non-ocean pixels that were at least 1 km from
- the interior of the land-ocean interface. We identified interior water pixels using GSW with at least

- 75% water occurrence from 1984-2019 and that were at least 0.0225 km<sup>2</sup> in area (to remove small
- lakes, ponds, and narrow streams). As a result, inland water bodies and the ocean-land interface are
- more distinct , more consistent, and better aligned.
- We summarized our estimates of human modification across all terrestrial lands, biomes, and
- ecoregions (defined by Dinerstein et al. 2017) and here report median  $(H_{median})$  and mean  $(H_{mean})$
- statistics. We summarized results of temporal trends using the mean annualized difference ( $H_{mad}$ ),
- calculated as the mean values across each analytical unit (e.g., biomes, ecoregions) of the annualized
- difference assuming a linear trend  $(H_{ad})$ :

$$H_{ad} = (H_u - H_t)/(u - t)$$
,

(3)

- where *u* and *t* are the years of the datasets (e.g., u=2015, t=1990) and *u*>t. When discussing trends
- between 1990 and 2015, we emphasize the mean statistic because it better captures locations where
- H values have changed (mostly increasing over time), partly due to land uses with high values (e.g.,
- urbanization ~0.8) that are not well represented in the median statistic. We calculated the increase in
- H, or conversely the amount of natural habitat loss, as the per-pixel value times the pixel area,
- summed across a given unit of analysis. This assumes that any increase in the level of human
- modification causes natural land loss regardless of the original *H* level. We also report the median
- statistic because, as is typical of spatial landscape data, the distribution of *H* values is skewed to the
- right. Finally, we compared our results of  $H_{mad}$  to those calculated on the Human Footprint (HF for
- 1993-2009; Venter et al. 2016) and the temporal human pressure index (THPI for 1995-2010; Geldmann
- et al. 2019b).

## 182 2.2 Stressors mapped

#### 183 2.2.1 Urban and built-up

- To map built-up areas that are typically found in urban areas and dominated by residential,
- commercial, and industrial land uses, we used the most recent version of the Built-up Grid from the
- Global Human Settlements Layers dataset (GHSL R2018A; Pesaresi et al. 2015). The degree of human
- modification that is contributed by built-up areas,  $H_{bu}$ , is:
- $H_{bu} = F_{bu} * p(C_{bu}) * I_{bu}$ ,

- (4)
- where  $F_{bu}$  measures the proportion of the area of a pixel classified as built-up,  $p(C_{bu})$  applies the
- GHSL-reported confidence mask (for 2014) for locations of the built-up areas (for the target year;
- Pesaresi et al. 2015) and  $I_{bu}$  is the intensity factor specified in Table 2.

#### 192 2.2.2 Agriculture

- We mapped agriculture stressors by identifying land cover classes associated with crop and
- pastureland from ESA CCI land cover datasets (ESA CCI 2015; Perez-Hoyos et al. 2017; Li et al. 2018)
- available at 0.09 km<sup>2</sup> for 1992, 2000, 2010, and 2015. We merged the cropland and pastureland
- stressors because these two classes are combined in the ESA land cover data, and they are
- challenging to distinguish even at higher resolution (~30 m, Wickham et al. 2017). To incorporate
- classification errors associated with all cover classes, we multiplied the footprint  $F_{cp}$  = 1.0 times the
- probability that a pixel with cover class C was found to be cropland or pasture,  $p(C_{cp})$ , by

- interpreting reported accuracy assessment results (ESA CCI 2017, in Table 3). To reduce the effects of 201 scattered pixels that have some probability of being mapped as cropland-pastureland (e.g.,
- misclassified pixels high-elevation tundra or alpine areas), we multiplied  $p(C_{cp})$  by the proportion of
- lands estimated to be in crops from the Unified Cropland Layer (Waldner et al. 2016), v so that:

$$204 \ p(C_{cp})' = p(C_{cp}) \times v , \qquad (5)$$

- and also reduced the value of  $p(C_{cp})$  based on patch size A, assuming that accuracy declines rapidly
- with cropland/pastureland small "patches" (A < 1 km<sup>2</sup>) using:

$$p(C_{cp})'' = (p(C_{cp})')^2$$
, A < 1. (6)

- We then calculated  $H_{cp}$  as: 209  $H_{cp} = F_{cp} * p(C_{cp})$ " \*  $I_{cp}$ .

We developed spatially-explicit estimates of agricultural intensity based on land management, such as cropping and number of rotations, tilling, and cutting operations, because these activities typically vary geographically (van asselen and Verburg 2012; Kehoe et al. 2017). We followed existing methods (Chaudhary and Brooks 2018) to estimate three intensities of agricultural land use -- minimal, light, and intense -- and then mapped them using cover types from Global Land Systems v2 dataset (GLS; Kehoe et al. 2017) by estimating intensity values (*I*) for each of the agricultural intensity types (Table

(7)

2). Although GLS v2 represents conditions circa 2005, we incorporated temporal changes by

weighting the proportions of agricultural lands from the time-varying ESA CCI land cover datasets.

- To estimate the modification associated with the grazing of domestic livestock ( $H_{au}$ ), we used the
- Gridded Livestock of the World v3 (Robinson et al. 2014; Gilbert et al. 2018a, Gilbert et al. 2018b) that
- maps the density of animals per  $km^2$  (G) for eight types of livestock (*j*): buffaloes, cattle, chickens,
- ducks, goats, horses, pigs, and sheep. To calculate the overall footprint of grazing ( $F_{au}$ ), we summed
- the weighted densities by global averages of livestock unit (*LU*) coefficients ( $w_i$ = 0.84, 0.67, 0.01, 0.01, 0.01, 0.04, 0.23, 0.10, listed respectively for each livestock species stated above). We used a
- lower threshold found at 10% to remove values <1.0 LUs/km<sup>2</sup> (similar to Jacobson et al. 2019) and
- 1000 LU km<sup>-2</sup> as an upper threshold because it is a common breakpoint between grazing and
- industrial feedlots (Gerber et al. 2010). We assumed (here, and below unless otherwise provided) no
- uncertainty ( $p(C_{au}) = 1.0$ ), because we lacked explicit data to do so. We then  $log_{10}$  transformed and
- max-normalized (Kennedy et al. 2019a) to obtain 0-1 values, and calculated the mean  $H_{au}$  using a 10 km radius moving window to reduce the effects of the coarser-resolution pixels:

$$F_{au} = \sum_{j=1}^{8} G_j w_j$$
, max(1000), min(1) (8)

$$H_{au} = \left( \left( \log \left( F_{au} + 1 \right) \right) / \log(1000) * p(C_{au}) * I_{au} \right).$$
(9)

#### 234 2.2.3 Energy and extractive resources

 $_{\rm 235}$  To estimate stressors associated with extractive energy production, we mapped gas flares derived

from "night-time lights" using data from the Visible Infrared Imaging Radiometer Suite from the

Suomi National Polar-orbiting Partnership (VIIRS; Elvidge et al. 2013). Roughly 90% of gas flares occur

- at locations where oil and gas are extracted (Elvidge et al. 2015). We used point data processed
- specifically to identify gas flares in VIIRS for 2012/2013 (Elvidge et al. 2016). For each flare, we

approximate the footprint of points (and lines) using a simple "buffer", which implicitly assumes no 241 location error and no distance-decay from the point of origin. Such a buffer approach essentially 242 centers a cylinder on each data point, where volume (V) equals the approximate footprint and height 243 (h) and a perfect certainty of 1.0. Here, however, we assumed some uncertainty in the location of the 244 point and that the effects associated with a feature such as an oil/gas well-head diminish with 245 distance. That is, rather than use a cylinder with volume V (or similarly a simple uniform buffer away 246 from linear features, e.g. powerlines or roads), we used a conic shaped kernel centered on the point 247 to calculate the uncertainty  $p(C_{og})$ , where the height of the cone h=0.5 represents a conservative 248 estimate of spatial accuracy (Theobald 2013). We derived the cone radius D = 0.329 km by setting V to 249 the footprint of 0.057 km<sup>2</sup>: 250  $D = \sqrt{(3/h) V/\pi}$ , 251 (10)

approximated a footprint of 0.057 km<sup>2</sup> per well head (Allred et al. 2015). It is common to

252 Thereby the uncertainty parameter for each point is calculated using:

$$p(C_{og}) = 3h/\pi D^2$$
.

(11)

(13)

We assigned the value of  $p(C_{og})$  that overlapped the center of each pixel, with max  $p(C_{og}) = 1.0$ .

Human modification was then calculated as:

$H_{og} = F_{og} * p(C_{og}) * I_{og}$  (12)

#### 257 2.2.5 Mines and quarries

To estimate modification due to mines and quarries, we derived locations represented as points from 258 a global mining dataset (n=34,565; S&P 2018; Valenta et al. 2019). We retained surface mines that 259 were constructed, construction started, in operation, in the process of being commissioned, or 260 residual production (n=22,705). For the temporal change analysis, we removed locations that did not 261 have a specified year of construction (n=3,634). We calculated the mean disturbed area and 262 associated infrastructure of a mine by intersecting mine point locations with 441,623 polygons that 263 represent footprints of quarries/mines (OpenStreetmap, 2016). For four types of mines: coal; 264 hard-rock (bauxite, cobalt, copper, gold, iron ore, lead, manganese, molybdenum, nickel, phosphate, 265 platinum, silver, tin, uranium oxide, and zinc); diamonds; and other (antimony, chromite, graphite, 266 ilmenite, lanthanides, lithium, niobium, palladium, tantalum, and tungsten), we estimated the mean 267 area (a) to be:  $12.95 \text{ km}^2(n=647)$  for coal,  $8.54 \text{ km}^2(n=860)$  for hard-rock,  $5.21 \text{ km}^2(n=39)$  for 268 diamonds, and 3.40 km<sup>2</sup> (n=27) for other. Finally, following equations 8 and 9, we calculated  $p(C_m)$  for 269 each of the four mining types using D of 4.973, 4.038, 2.548, and 3.154 km, respectively, and 270

calculated
$$H_m$$
 as:

$$H_m = F_m * p(C_m) * I_m$$
.

#### 273 2.2.6 Power plants

- To estimate the effects of where energy is produced, we mapped the location of power plants
- represented as points (n=29,903; WRI 2019). For the temporal change analysis, we removed locations
- that did not have a specified year of construction (n=16,288). We estimated  $p(C_{pp})$  using a
- conic-shaped kernel (Eqs. 8 and 9) and h=0.5. We mapped both non-renewable energy forms ( $H_{ppn}$ ;
- coal, oil, natural gas) and renewable energy forms ( $H_{ppr}$ ; geothermal, hydro, solar, wind), where we

assumed  $F_{pp}$ =1 and calculated a single  $p(C_{pp})$  for both non-renewable and renewable energy sectors with  $D_{pp}$ =1224 m (following Theobald 2013):

$$H_{ppn} = F_{pp} * p(C_{ppn}) * I_{ppn}$$
, (14)  
$H_{ppr} = F_{pp} * p(C_{ppr}) * I_{ppr}$ . (15)

#### 283 2.2.7 Transportation and service corridors

For transportation, we mapped roads and railways using OpenStreetMap highway linear features (OpenStreetMap, 2019). We calculated the footprint for the following transportation types: major (motorway, primary, secondary, trunk, link), minor (residential, tertiary, tertiary-link), two-track

roads and railways as:

$$F_{rr} = \sum_{i=0}^{\infty} (w / \alpha) * \mu$$
, (16)  
$H_{rr} = F_{rr} * p(C_{rr}) * I_{rr}$ , (17)

where w is the estimated width of a road of type *i* from Table 2,  $\alpha$  is the pixel width (i.e. 300 m), and  $\mu$ =0.79 to adjust for the fractal dimension of road lines crossing cells (Theobald 2000) because road lines often cross pixels at random angles. If a divided highway is represented as two separate lines, then each is represented independently. Also, if a cell has two or more roadway types cross it (e.g., where a secondary road joins a highway), the fuzzy sum of  $H_{rr}$  for both roads is calculated. Note that

- use of roads is incorporated into the "human intrusion" stressor (described below).

To map the modification associated with above-ground powerlines  $(H_{n})$ , we used:

$$H_{pl} = F_{pl} * p(C_{pl}) * I_{pl}$$
,

where  $F_{pl}$  is calculated using a 500 m buffer (Theobald 2013), and  $p(C_{pl})$  is calculated using h=0.5, and 300  $I_{pl}$  is the estimate of intensity.

(18)

- To estimate a stressor associated with electrical infrastructure and energy use  $(H_n)$ , we mapped

"night-time lights" using the Defense Meteorological Satellite Program/Operational Linescan System

(DMSP/OLS; Elvidge *et al.*, 2001) "stable" lights dataset. We included this as a distinct stressor from

the energy extraction stressor (oil and gas flares, discussed above) because gas flares are derived by

finding anomalies (high values) in the images rather than from the "stable lights" product, and the

footprints associated with the flares are an extremely small fraction of the overall extent of energy 308 infrastructure.

To maximize temporal consistency, we used the intercalibrated DMSP/OLS dataset (Zhang et al. 2016; Li and Zhou 2017) and extended their approach for 2013 (using *a*=1.01, *b*=0.00882, *c*=-0.965; Zhang et al. 2016). DMSP/OLS values, *L*, are expected to range from 0 to 63, but because max values differed yearly (ranging from 57.87 - 66.16), we normalized all images (1992-2013) to range from 0 to 1.0 using the max-adjusted value for each year (*L*'). To reduce the effects of noise in the images in areas with low-light and in high northerly latitudes, we removed nighttime light values when *L*'<0.077 -- that is, we set values to *null* when they were below the 25<sup>th</sup> percentile of the global terrestrial distribution compared to the often used noise threshold of *L*=5 (following Elvidge et al. 2001).

- To adjust for inter-annual spatial-misalignment errors (Elvidge et al. 2013), we adjusted the
- normalized DMSP image for 2013 to align with the 2013 VIIRS product by identifying sharply 320
- contrasting and consistent signals at 10 locations (n=10) distributed across the continents. We then 321
- visually compared each of the images from 1992-2012 to the DMSP image for 2013 and shifted the 322 images to align them (averaged shift in meters: x=359.5, y=476.2). To further reduce inter-annual 323
- variability, we averaged image values at each pixel using a 3-year "tail" and used a 324
- rank-ordered-centroid weighting (Roszkowska 2013) such that the spatially-aligned and 325
- temporally-smoothed nightlight value Y for year t is: 326
- $Y_{t} = (L_{t}^{'} * 0.62) + (L_{t-1}^{'} * 0.26) + (L_{t-2}^{'} * 0.12)$ . 327 (19) 328
- Finally, to reduce the blooming effects and to take advantage of the higher-quality VIIRS-based
- nightlights (i.e. higher spatial resolution, reduction of saturated pixels), we sharpened DMSP
- nightlight values y, using the VIIRS brightness value y to be proportional to the ratio of the DMSP 331 332 values:
- $Y'_t = Y_t * (L'_t \div L_{2013})$ . (20)
- We then transformed  $Y_t$  following Kennedy et al. (2019a), capping values above 126.0 (the 99.5 335 percentile of global values):

$$H_{nl} = \left( \log_{10} \left( 1 + Y_{t}^{'} \right) / 2.104 \right) * p(C_{nl}) * I_{nl}$$
 (21)

- 2.2.8 Logging
- To estimate stressors on forested lands, we used maps of forest loss (Curtis et al. 2018) associated 339 with commodity-driven deforestation, shifting agriculture, and forestry. (Note that we excluded 340 wildfire as a stressor because of the challenges of attributing wildfires to human causation--341 especially over global extent, and urbanization because it is measured directly by the built-up 342 stressor). We then identified locations where forest was lost due to one of the three mapped 343 stressors (using v1.6, updated to 2018; Hansen et al. 2013) prior to the year of our estimated human
- modification map, and applied the intensity value associated with that stressor. Thus, (22)

$$H_{fr} = F_{fr} * p(C_{fr}) * I_{fr}$$
 ,

where  $F_{fr}$  is pixels of forested loss in a given year, and  $I_{fr}$  is an estimate of intensity associated with 347 the cause of forest loss.

#### 348 2.2.9 Human intrusion

- We estimated human intrusion ( $H_i$ ) using a method that builds on and extends accessibility modeling
- (Nelson 2008; Theobald 2008, 2013; Theobald et al. 2010; Weiss et al. 2018; Nelson et al. 2019). Human
- intrusion (aka "use": Theobald 2008) uses central place theory (Alonso 1960) and integrates human 351
- accessibility throughout a landscape from defined locations, typically along roads and rails as well as
- off-road areas from urban areas (Theobald et al. 2010; Esteves et al. 2011; Theobald 2013; Larson et al. 353
- 2018).
- Accessibility measured in travel time in minutes is calculated from each mapped settlement point j
- (e.g., cities, towns, villages) from GRUMP v1.01 and GPW v4 (CIESIN 2017, 2018). This approach is 357
- much less sensitive to arbitrary thresholds of city/town size (e.g., 50,000 residents), often used due

to computational constraints (e.g. Nelson 2008; Weiss et al. 2018). Second, to estimate "intrusion" of people to adjacent areas from a given settlement, we estimated the number of people (using 360 population estimates at settlement j) at a given location (X; ~population density: people/km<sup>2</sup>) 361 362 following the assumption that the human density halved with every 60 minutes traveled (Theobald 2008, 2013). The resulting intrusion map for each settlement was then summed to account for typical 363 overlaps of intrusion from nearby settlements. We assumed that there is a limit at very high 364 population densities, and so we capped the maximum value of intrusion, X, at 1,000,000 then 365

max-normalized using a square-root transform: 366

$$F_i = X^{0.5} * 0.001$$
, (23)  
$H_i = F_i * p(C_i) * I_i$ . (24)

$$H_i = F_i * p(C_i) * I_i$$
.

Note that accessibility was calculated using estimates of travel time along roads and rails, as well as

off-road through different features of the landscape, using established travel time factors (Tobler 371

1991) and presuming walking off-trail or via boats on freshwater or along ocean shoreline (Nelson 372

2008; Theobald et al. 2010; Weiss et al. 2018; Nelson et al. 2019). This included effects of international 373

borders following Weiss et al. (2018), and accessibility to lands was calculated across oceans. 374

#### 2.2.10 Natural systems modification 375

Dams and their associated reservoirs flood natural habitat and strongly impact the natural flow 376

regimes of the adjacent rivers (Grill et al. 2019). We mapped the footprint of reservoirs F, created 377

from 6,849 dams from the Global Reservoirs and Dams database (GRanD v1.3; Lehner et al. 2011;

http://globaldamwatch.org/grand/). 379

 $H_r = F_r * p(C_r) * I_r \quad .$ (25)380

Because there are some potential analyses that would benefit from treating all water bodies 382

consistently, we provided an additional version with all water bodies masked out in the dataset. 383

#### 384 2.2.11 Pollution

- We estimated the stress of air pollution by using data on nitrogen oxides (NO<sub>2</sub>) through time from
- the Emissions Database for Global Atmospheric Research (EDGAR v4.3.2; Crippa et al. 2018). We
- selected NO, because it is a strong contributor to acid rain/fog and tropospheric ozone and because 387
- atmospheric levels are predominantly from human-sources (Delmas et al. 1997). We used the 99th 388
- percentile (46,750 M tonnes) as the maximum value and then max-normalized ( $F_{nax}$ ) and adjusted 389

using the intensity value  $I_{nox}$ :

$$H_{nox} = F_{nox} * p(C_{nox}) * I_{nox}$$
 (26)

## 392 2.3 Uncertainty and validation analyses

To understand the uncertainty of our results associated with our estimated intensity values (Table 2),

following Kennedy et al. (2019b), we re-calculated H where I, was randomized (n=50) between the

minimum and maximum intensity values for each stressor. We then calculated the per-pixel mean

and standard deviation for the 50 randomizations at 1 km<sup>2</sup> resolution for computational efficiency

- and provide corresponding maps. 397

We also assessed the accuracy of our maps following validation procedures described in Kennedy et al. (2019a, 2019b, 2019c). Because historical "ground truth" human modification data in comparable 400 form are not widely available, we restricted our analysis to test the contemporary conditions of 401 human modification (~2017 map) that included all stressor layers. We used an independent validation 402 dataset from Kennedy et al. (2019a) that quantified the degree of human modification from visual 403 interpretation of high resolution aerial or satellite imagery across the world. We selected plots using 404 the Global Grid sampling design (Theobald 2016), a spatially-balanced and probability-based random 405 sampling that was stratified on a five-class rural to urban gradient using "stable nighttime-lights" 406 2013 imagery (Elvidge et al., 2001). Within each of 1,000 ~1 km<sup>2</sup> plots, we selected 10 simple-random 407 408 locations to capture rare features and heterogeneity in land use and land cover (for a total of 10,000 sub-plots), which were separated by a minimum distance of 100 m. The spatial-balanced nature of 409 410 the design maximizes statistical information extracted from each plot, because it increases the number of samples in relatively rare areas that are likely of interest (in contrast to simple random 411 sampling) -- especially for urbanized and growing cities (Theobald, 2016).

## 413 2.4 Processing platform

- We processed, modeled, and analyzed the spatial data using the Google Earth Engine platform
- (Gorelick et al. 2017). We calculated all distances and areas using geodesic algorithms in decimal
- degrees (EPSG: 4326). We summarized areas and percentages after projecting the data to Mollweide 416
- equal-area (WGS84) to simplify calculations. All datasets and maps conform to the Google Earth 417
- Engine terms of service. We used program R 3.6.1 (R Core Team 2019) to generate Fig. 2. 418

# 419 3 Results

- Below we describe the temporal and spatial trends of human modification by continents (Table 4),
- biomes (Table 5), and ecoregions (Fig. 2). 421

## 422 3.1 Changes from 1990-2015

- The mean value of H for global terrestrial lands increased from 0.0822 in 1990 to 0.0946 in 2015, a
- percentage change of 15.04% overall and 0.60% annually (Table 4). This equates to 1.6 M km<sup>2</sup> of
- natural lands lost -- 178 km<sup>2</sup> daily. Increases in human modification occurred globally and in both 425
- urban and rural locations. We found that the largest increases in H<sub>mad</sub> occurred in Oceania, followed 426
- by Asia and Europe. Australia had the lowest increase followed by North and South America (Table
- 4). The biomes that exhibited the greatest increases in modification were mangroves; tropical & 428
- subtropical moist broadleaf forests; and tropical & subtropical dry broadleaf forests; while the 429
- biomes with the smallest increases were tundra; boreal forests/taiga; and deserts and xeric 430
- shrublands. Maps of changes in H<sub>mad</sub> between 2015 and 1990 for each ecoregion are shown in Fig. 1a, 431
- relative to HF (Fig. 1b) and THPI (Fig. 1c). Figure 2 shows the ratio of natural land loss between 1990

and 2015, for each ecoregion and grouped by biome, in the context of the contemporary extent of

human modification. We found most ecoregions (n=814) had increased in human modification, while

the few (n=32) that had decreased were concentrated in higher latitudes and in more remote areas.

We also found that changes in  $H_{mad}$  have increased over time, from 0.0004 to 0.0005 to 0.0006,

during 1990-2000 to 2000-2010 to 2010-2015. The percent change has also increased over time from

0.51% to 0.59% to 0.68%.

## 439 3.2 Contemporary extent

We found that about 19.1 M km<sup>2</sup> (±0.0013) of natural lands were lost by ~2017 -- about 14.6% of land

globally (Table 4). South America was the most transformed (28.7%), followed by North America

(16.8%), while Australia (5.0%) and Africa (10.7%) were the least transformed. Broad-scale patterns of

the extent of human modification in ~2017 are shown in Fig. 3. Note that "natural lands lost" was

calculated using the continuous value of *H*, rather than approximations based on classifying the

445 distribution.

Terrestrial lands with very low levels of human modification (H<0.01; Kennedy 2009c, Riggio et al.

- 2020) are concentrated in less productive and more remote areas in high latitudes and dominated by
- inaccessible permanent rock and ice or within tundra, boreal forests, desert regions, and to a lesser
- extent montane grasslands. Table 5 shows that the biomes with the highest levels of H in ~2017 were
- temperate broadleaf and mixed forests (H=0.3744); tropical & subtropical dry broadleaf forests
- (H=0.3317); and Mediterranean forests, woodlands & scrub (H=0.2903). The five least modified

biomes were tundra (mean H=0.0023); boreal forests/taiga (H=0.0213); deserts and xeric shrublands

- (H=0.0572); and montane grasslands and shrublands (H=0.0894).

Following thresholds from Kennedy et al. (2019a), we found that in ~2017, 51.0% of global lands had a

- mean value  $H \le 0.01$  (i.e. very low human modification, ), 13.3% had a mean of  $0.01 < H \le 0.1$  (low),
- 21.0% had a mean of 0.1 <  $H \le$  0.4 (i.e. moderate), 12.3% had a mean value of 0.4 <  $H \le$  0.7 (high), and
- 2.4% (3.2 M km<sup>2</sup>, ±0.0003) had a mean of 0.7 <  $H \le 1.0$  (very high). By area, these results by class
- amount to: very low=66.8 M km<sup>2</sup> ( $\pm$ 0.0067), low=17.4 M km<sup>2</sup> ( $\pm$ 0.0017), moderate= 27.6 M km<sup>2</sup>
- (±0.0028), high=16.1 M km<sup>2</sup> (±0.0016), and very high=3.18 M km<sup>2</sup>, ±0.0003). Also, we found that 17.6%
- had a mean value of H < 0.0001 (23.0 M km<sup>2</sup>, ±0.0023), and 4.2% had H < 0.00001 (5.5 M km<sup>2</sup>,
- ±0.0006).

## 464 3.3 Comparisons

We compared our work to earlier efforts (summarized in Table 6) to determine if overall trends and

extents were generally consistent and with similar priorities of biomes and ecoregions. Globally,  $H_{mad}$

- from 1990-2015 (t=1990, u=2015) was 0.0005, while for HF and THPI it was higher ( $HF_{mad}$ =0.0006,
- THPI<sub>mad</sub>=0.0008). Perhaps more important is that the variability of the mean annualized difference
- values in the HF and THPI was 2.3 and 3.2 times that of H. By continent, we found that  $H_{mad}$  increased
- the most in Oceania, followed by Asia, Europe, Africa, South America, North America, and Australia.
- Continental ranks by THPI followed H roughly, though HF differed more substantially (Table 5).  $H_{mad}$

increased for all continents, but HF<sub>mad</sub> showed *declines* in modification for Europe and South America,
while THPI<sub>mad</sub> showed a decline for North America.

- We also found the ranking of biomes by mean annualized difference for HF and THPI were fairly
- different from ranks developed from H values (Table 7). Of the three biomes with the largest increase
- for H<sub>mad</sub>, two were also identified by HF (tropical & subtropical dry broadleaf forests and tropical &
- subtropical moist broadleaf forests) and none THPI. Of the five biomes with the largest increase for
- $H_{mad}$ , three were also identified by HF and THPI. The biomes that had the greatest disagreement
- amongst the ranking of H, HF, and THPI were mangroves; tropical & subtropical coniferous forests;
- and tropical & subtropical dry broadleaf forests.
- The biggest differences in rankings between the H and the HF were for temperate and broadleaf mixed forests (and see comparisons of H1k and HF in Kennedy et al. 2019a, 2019b, Riggio et al. 2020). 484 HF was estimated to result in 12.3% modification for an earlier date (~2009; Venter et al. 2016) and is 485 lower likely because fewer stressors were included, its additive combination method, and its strongly 486 right-skewed distribution caused by max-value normalization. The ranks of the extent of modification 487 by biomes, however, were very similar between H, H1k, and HF. In general, H had intermediate 488 modification levels compared to H1k and HF: with H1k levels being slightly higher (difference 489 between 0.00 min to 0.09 and average difference of 0.05 by biome) and HF being slightly lower 490 (difference between 0.00 min to 0.13 max and average difference of 0.04 by biome; Table 7). Results 491 492 for ecoregions shown in Fig. 1 are even more striking, as the mean annualized difference values for
- HF and THPI were inconsistent with our results. Of the 814 ecoregions that had increases in  $H_{mad}$ , a
- decrease in modification was found for 201 ecoregions in  $HF_{mad}$  and 202 for  $THPI_{mad}$ ; and for the 32
- ecoregions that were found to have decreases in  $H_{mad}$ , an increase in modification was found for 20 in
- HF and 22 in THPI.
- Finally, the global estimate for H1k was likely higher than H because H1k did not limit the livestock
- stressor at LU km<sup>-2</sup> <1.0, used a slightly higher value for the low-threshold on the electrical
- infrastructure and energy use stressor (i.e. "nightlights"), and reported results that incorporate
- uncertainty in estimates of intensity. Furthermore, global modification from farming was estimated
- at 37% for 2000 (Ramankutty et al. 2008) compared to 14.6% with H. The difference with our results is
- <sup>503</sup> largely due to their mapping of the area land cover types but not differentiating the intensity of the
- impact of those cover types (crop and pasture).

## 505 3.4 Uncertainty and validation analyses

- We addressed uncertainty in our results by incorporating the parameter  $p(C_{s})$  for every sector s to
- best quantify uncertainty in its spatial location and classification as detailed in section 2.2.; for
- example, we adjusted  $p(C_{cn})$  by directly incorporating measured confusion among land cover types
- using the results from the accuracy assessment of the land cover dataset (from Eq. 4). Additionally,
- we incorporated uncertainty by calculating the global mean for each of the 50 randomizations, which
- across the 50 iterations was 0.1434 (SD= ±0.0076) and ranged from 0.1243 to 0.1612. Thus, the global
- mean of 0.1461 obtained using our "best-estimate" intensity values was in line with our uncertainty

<sup>513</sup> results. We also mapped the per-pixel variance (standard deviation) to examine the spatial pattern of

uncertainty (Figure 4). The locations of the highest levels of uncertainty tend to be in more highly

developed landscapes.

We found strong agreement between *H* for ~2017 and our validation data (r=0.783), with an average 518 root-mean-square-error of 0.22 and a mean-absolute-error of 0.04, for the 926 ~1 km<sup>2</sup> plots (9,260 519 sub-plots). There were 726 plots within ±20% agreement, while for 161 plots *H* was estimated higher 520 than our visual estimate from the validation data (and 39 plots lower). Estimates of *H* were biased 521 high, likely because the stressors for the "human intrusion" and electrical infrastructure (based on 522 nighttime lights) are not readily observable from the aerial imagery used to generate the validation 523 data. Our results here are consistent with our earlier findings (Kennedy et al. 2019a, 2019b, 2019c).

# 524 4 Discussion

## 525 4.1 Summary

We found rapid and increasing human modification of terrestrial systems, resulting in the loss of 526 natural lands globally. Our findings foreshadow trends and patterns of increased human 527 modification, assuming future trends in the next 25-30 years continue as they have recently. Thus, 528 our study reinforces calls for stronger commitments to help reduce habitat loss and fragmentation 529 (Kennedy et al. 2019a, Jacobson et al. 2019) -- which should be considered in conjunction with current 530 commitments to reduce CO, emissions through the Paris climate accord (Baruch-Mordo et al. 2019; 531 Kiesecker et al. 2019). We believe that the comparisons of ecoregions and biomes offer valuable 532 contextual information that provides initial guidance on conservation strategies that may be most 533 appropriate (Kennedy et al. 2019a). Also, it is important to consider the feasibility of achieving 534 ecoregional (Dinerstein et al. 2017) or ecosystem (Jantke et al. 2019) representation goals, as well as 535 additional stresses caused by climate change (Costanza and Terando 2019). We emphasize that 536 although global, continental, biome, and ecoregional summaries provide a general understanding of 537 trends and patterns, the high resolution of H and its gradient nature supports robust estimates of 538 change in human modification within a country and within an ecoregion, which are essential for 539 tracking progress toward international and national conservation commitments (Mace et al. 2018), 540 especially when placed within a broader structured decision-making framework (Tullock et al. 2015) 541 542 Our datasets of human modification provide the most granular, contemporary, comprehensive, 543 high-quality, and robust data currently available to assess temporal and spatial trends of global 544 human impacts on landscapes. Our work is grounded in a structured classification of stressors, uses 545 an internally-consistent model, evaluates uncertainty, and incorporates refinements to minimize the 546 effects of scaling and classification errors. Our validation approach uses an independent and 547 spatially-balanced random sample design to provide strong support for the quality of our findings

spatially-balanced random sample design to provide strong support for the quality(Kennedy et al. 2019c).

Our overarching goal in producing and publishing these datasets is to support detailed quantification 551 of the rates and trends, as well as the current extent and pattern, and to understand the degree of 552 human modification across the continuum from low (e.g., wilderness) to high (e.g., intense urban). 553 Beyond the basic findings presented here, we believe there are many potential applications of these 554 datasets, including: examining temporal rates and trends of land modification in and around 555 protected areas (e.g., Geldmann et al. 2019a); estimating fragmentation for ecoregions and biomes 556 (Kennedy et al. 2019a, Jacobson et al. 2019); and evaluating conservation opportunities and risks 557 (e.g., the conservation risk index; Hoekstra et al. 2005). We also note that the human modification 558 approach allows, in a straightforward and logically consistent way, inclusion of additional stressors 559 and higher resolution datasets that may become available over time or may be available for specific, 560 local areas. 561

#### 562 4.2 Caveats

As with any model, we recognize there are limitations of our work. We did not include data for all human stressors, largely because of incomplete global coverage or coarse mapping units (Klein 564 Goldewijk et al. 2007; Geldmann et al., 2014), an inability to discern human-induced versus natural 565 disturbances, or uncertainty in the location and directionality of its impact (e.g.; climate change on 566 terrestrial systems; Geldmann et al., 2014). In particular and discussed in Kennedy et al. (2019a, 567 2019b), changes to land cover due to ecological disturbance events, such as wildfires or flooding, are 568 not included in our analysis because of the difficulty in separating natural from human-caused 569 disturbances -- yet, we recognize that the broad extent of wildfire in particular, could have strong 570 implications. We did not include climate data as a stressor in this product to keep our analysis 571 manageable and tractable. Although we attempted to map each stressor comprehensively, we 572 recognize that some datasets may have missing features, particularly for mine and oil/gas wells --573 though large mines and concentrated oil/gas fields have been mapped quite well. For more 574 integrated analyses, our data product should be used in combination with datasets of impacts due to 575 climate change (e.g., Parks et al. 2020). 576 577 578 Stressors that are particularly important to improve include effects of grazing (currently coarse data and very broad expanse), pasture land, invasive species, and climate change (especially wildfire and 579 effects of sea-level rise), and we encourage future work to focus on developing appropriate datasets 580 and approaches to include or better capture these stressors. Key datasets we believe should be 581 improved include transportation networks, including logging roads (e.g., Van Etten 2019) that are 582 comparable through time; livestock grazing, rangelands, croplands, timber plantations, and 583

pasturelands and their intensity of use; resource extraction (especially mining footprints); and

585 temporal trends in gas flares, utility-scale solar and wind installations (Dunnett et al. 2020), and

586 electrical substations.

## 587 4.3 Data availability

The datasets generated from this work are available at <u>https://zenodo.org/record/3963013</u> (Theobald

et al. 2020), which includes the land/water mask used to support subsequent analyses. Extracts of

specific geographic areas can be obtained by contacting the authors. All other datasets used in our

work are open-source data cited within.

# 592 Author contributions

DT, CK, BC, JO, SBM, JK conceived the paper; DT, CK, JO, BC prepared data; DT implemented the

model; DT, CK, BC, SBM conducted summary analyses; DT, CK, BC, JO, SBM, JK developed

recommendations; all contributed to writing the manuscript.

Competing interests: The authors declare that they have no conflict of interest.

# 598 Acknowledgments

We thank OpenStreetMap contributors (copyright OpenStreetMap and data are available from 600 https://www.openstreetmap.org), and E. Lebre for assistance with the global mining data.

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

# 872 Author information

David M. Theobald<sup>1,2</sup>, Christina Kennedy<sup>3</sup>, Bin Chen<sup>4</sup>, James Oakleaf<sup>3</sup>, Sharon Baruch-Mordo<sup>3</sup>, Joe

Kiesecker<sup>3</sup>

<sup>1</sup>Conservation Planning Technologies, Fort Collins, CO 80521, USA

- <sup>876</sup> <sup>2</sup>Department of Fish, Wildlife, and Conservation Biology, Colorado State University, Fort Collins, CO
- 80523, USA
- <sup>3</sup>Global Lands Program, The Nature Conservancy, Fort Collins, CO 80524, USA
- <sup>4</sup>Department of Land, Air and Water Resources, University of California, Davis, CA 95616, USA

# 881 Tables

Table 1. Overview of stressors, datasets, spatial resolution, and years data were available and used in

the maps of human modification. Stressor classification levels in parentheses correspond to those

within the Direct Threats Classification v2 (Salafsky et al. 2008). Acronyms of source data are bolded

in the Source column for reference throughout the paper. For each stressor, the years 1990-2015 are

used for change analysis, and ~2017 is a compilation of all stressors that represents "current"

conditions with the median year of 2017.

| Class Stressor* Source                  |                                                                                                                                                                                                                                                   | Source                                                                                                                                                                                                                                 | Scale           | Year  |      |                      |      |                      |  |
|-----------------------------------------|---------------------------------------------------------------------------------------------------------------------------------------------------------------------------------------------------------------------------------------------------|----------------------------------------------------------------------------------------------------------------------------------------------------------------------------------------------------------------------------------------|-----------------|-------|------|----------------------|------|----------------------|--|
|                                         |                                                                                                                                                                                                                                                   |                                                                                                                                                                                                                                        | (km²)           | 1990  | 2000 | 2010                 | 2015 | ~2017                |  |
| Urban &<br>built-up (1)                 | Built-up (1.1,<br>1.2)                                                                                                                                                                                                                            | Global Human Settlement<br>Layer version R2018A<br>( <b>GHSL;</b> Pesaresi et al. 2015)                                                                                                                                                | 0.0009<br>- 0.9 | 1990  | 2000 | 2010 <sup>*</sup>    | 2015 | 2015                 |  |
| Agriculture<br>(2)                      | Croplands &<br>pasturelands<br>(2.1)                                                                                                                                                                                                              | European Space Agency<br>Climate Change Initiative<br>land cover ( <b>ESA CCI</b> ; Li et<br>al. 2018)<br>Unified Cropland Layer<br>( <b>UCL</b> ; Waldner et al. 2016)<br>Global Land Systems v2<br>( <b>GLS</b> ; Kehoe et al. 2017) | 0.9<br>1<br>1   | 1992  | 2000 | 2010<br>2010<br>2010 | 2015 | 2015<br>2010<br>2010 |  |
| Grazing (2.3                            |                                                                                                                                                                                                                                                   | Gridded Livestock of the<br>World v3 ( <b>GLW</b> ; Robinson<br>et al. 2014; Gilbert et al.<br>2018a, Gilbert et al. 2018b)                                                                                                            | 10              |       |      | 2010                 |      | 2010                 |  |
| Energy<br>production<br>& mining<br>(3) | Oil & gas<br>production<br>(3.1)Nighttime flares from<br>Defense Meteorological<br>Program/Operational<br>Line-scan System<br>(DMSP/OLS, Elvidge et al.<br>2009) and Visible Infrared<br>Imaging Radiometer Suite<br>(VIIRS, Elvidge et al. 2016) |                                                                                                                                                                                                                                        | 0.25 -<br>1.0   |       |      |                      |      | 2016                 |  |
|                                         | Mining &<br>quarrying<br>(3.2)                                                                                                                                                                                                                    | S&P global mining dataset<br>(S&P 2018; Valenta et al.<br>2019)                                                                                                                                                                        | ~1:100<br>00    | 1990, | 2000 | 2010                 | 2015 | 2018                 |  |
|                                         | Renewable<br>(3.3) &<br>non-renewab                                                                                                                                                                                                               | World Resources Institute<br>Power plants ( <b>WRI;</b> WRI<br>2019)                                                                                                                                                                   | ~1:100<br>000   | 1990  | 2000 | 2010                 | 2015 | 2018                 |  |

|                                              | le power (1.2)<br>generation                   |                                                                                                                                                                                                                                                        |                 |       |      |      |      |      |
|----------------------------------------------|------------------------------------------------|--------------------------------------------------------------------------------------------------------------------------------------------------------------------------------------------------------------------------------------------------------|-----------------|-------|------|------|------|------|
| Transporta<br>tion &<br>service<br>corridors | Roads (4.1)                                    | OpenStreetMap highway,<br>minor, and two-track<br>features ( <b>OSM</b> ;<br>OpenStreetMap 2019)                                                                                                                                                       | ~1:10-2<br>5000 | **    |      |      | **   | 2019 |
| (4)                                          | Railways<br>(4.1)                              | OSM railway features<br>(OpenStreetMap 2019)                                                                                                                                                                                                           | ~1:10-2<br>5000 |       |      |      |      | 2019 |
|                                              | Powerlines<br>(4.2)                            | OSM power line features<br>(OpenStreetMap 2019)                                                                                                                                                                                                        | ~1:10-2<br>5000 |       |      |      |      | 2019 |
|                                              | Electrical<br>infrastructur<br>e (4.2)         | Nighttime lights from<br><b>DMSP/OLS</b> and <b>VIIRS</b><br>(Elvidge et al. 2001; Doll<br>2008; Elvidge et al. 2013;<br>Zhang et al. 2016)                                                                                                            | 0.25 -<br>1.0   | 1992  | 2000 | 2010 | 2015 | 2018 |
| Biological<br>harvesting<br>(5)              | Logging &<br>wood<br>harvesting<br>(5.3)       | Forest loss (Curtis et al.<br>2018) and forest change<br>(Hansen et al. 2013)                                                                                                                                                                          | 0.09 -<br>100   | 2000  | 2000 | 2010 | 2015 | 2018 |
| Human<br>intrusions<br>(6)                   | Human<br>intrusions<br>(1.3, 5.1, 5.2,<br>6.1) | Human intrusion<br>(Theobald 2008, <b>HUE</b> )<br>using accessibility and<br>population from Global<br>Rural-Urban Mapping<br>Project v1.01 ( <b>GRUMP</b> ;<br>CIESIN 2017) and Gridded<br>Population of the World v4<br>( <b>GPW</b> ; CIESIN 2018) | 1               | 1990* | 2000 | 2010 | 2015 | 2015 |
| Natural<br>system<br>modificatio<br>ns (7)   | Reservoirs<br>(7.2)                            | Global Reservoirs and<br>Dams ( <b>GRanD</b> v1.3; Lehner<br>et al. 2011)                                                                                                                                                                              | ~1:250<br>00    | 1990  | 2000 | 2010 | 2015 | 2017 |
| Pollution<br>(9)                             | Air pollution<br>(9.5)                         | Emissions Database for<br>Global Atmospheric<br>Research ( <b>EDGAR</b> v4.3.2;<br>Crippa et al. 2018) for NO <sub>x</sub>                                                                                                                             | ~100            | 1990  | 2000 | 2010 | 2012 | 2012 |

\*Based on interpolation. \*\*Used major roads (i.e. highways) for 2019.

Table 2. Estimates of the intensity value for each stressor. "Best" estimates were determined from

Brown and Vivas (2005)<sup>1</sup>, Theobald (2013)<sup>2</sup>, Kennedy et al. (2019a)<sup>3</sup>, or expert judgement<sup>4</sup>, and are

- $\,_{892}\,$  bracketed by a minimum and maximum range, following the lowest-highest-best estimate elicitation
- procedure to reduce bias (McBride et al., 2012). Results presented here use the best estimate, while
- minimum and maximum estimates are used to specify the range of possible randomized intensity
- values in the uncertainty analysis.

| Class                               | Stressor                                                                                                                                               | Minimum              | Best                 | Maximum              |
|-------------------------------------|--------------------------------------------------------------------------------------------------------------------------------------------------------|----------------------|----------------------|----------------------|
| Urban & built-up                    | Built-up areas <sup>3,4</sup>                                                                                                                          | 0.69                 | 0.85                 | 1.00                 |
| Agriculture                         | Cropland/pasture <sup>3</sup><br>- Minimal <sup>4</sup><br>- Light <sup>4</sup><br>- Intense <sup>1,4</sup>                                            | 0.29<br>0.35<br>0.60 | 0.34<br>0.45<br>0.65 | 0.39<br>0.55<br>0.70 |
|                                     | Livestock grazing <sup>1</sup>                                                                                                                         | 0.20                 | 0.28                 | 0.37                 |
| Energy production                   | Oil & gas production <sup>1,3</sup>                                                                                                                    | 0.70                 | 0.85                 | 1.00                 |
| & mining                            | Mining <sup>3</sup>                                                                                                                                    | 0.83                 | 0.91                 | 1.00                 |
|                                     | Power generation <sup>1</sup><br>(non-renewable)                                                                                                       | 0.70                 | 0.85                 | 1.00                 |
|                                     | Power generation (renewable) <sup>1</sup>                                                                                                              | 0.70                 | 0.80                 | 0.90                 |
| Transportation & service corridors* | Major roads <sup>1</sup>                                                                                                                               | 0.78<br>(20)         | 0.80<br>(30)         | 0.83<br>(40)         |
|                                     | Minor roads <sup>1</sup>                                                                                                                               | 0.39<br>(15)         | 0.44<br>(20)         | 0.50<br>(25)         |
|                                     | Two-track roads <sup>1</sup>                                                                                                                           | 0.10<br>(3)          | 0.15<br>(5)          | 0.20<br>(10)         |
|                                     | Railways <sup>1</sup>                                                                                                                                  | 0.78<br>(15)         | 0.80<br>(20)         | 0.83<br>(25)         |
|                                     | Powerlines <sup>2</sup>                                                                                                                                | 0.10                 | 0.15                 | 0.20                 |
|                                     | Electrical infrastructure (night-time lights) <sup>3</sup>                                                                                             | 0.20                 | 0.35                 | 0.50                 |
| Biological<br>harvesting            | Logging & wood harvesting <sup>1,4</sup> **<br>- Commodity-driven <sup>1,4</sup><br>- Shifting agriculture <sup>1,4</sup><br>- Forestry <sup>1,4</sup> | 0.60<br>0.10<br>0.10 | 0.65<br>0.20<br>0.20 | 0.07<br>0.30<br>0.30 |

| Human intrusion                 | Human intrusion <sup>3,4</sup>   | 0.20 | 0.35 | 0.50 |
|---------------------------------|----------------------------------|------|------|------|
| Natural systems<br>modification | Reservoirs <sup>4</sup>          | 0.60 | 0.65 | 0.70 |
| Pollution                       | Air pollution <sup>4,</sup> **** | 0.05 | 0.10 | 0.20 |

\*Assumed width of roads and railways (meters) provided in parentheses. Use of roads is

897 incorporated into estimates of human "intrusion".

\*\*Causes of forest loss due to wildfire was not included because of the challenges in understanding

human-causation/suppression, especially over a global extent. Also, cause of loss due to urbanization

was not included in this stressor because it is incorporated directly in the built-up stressor.

\*\*\*Minimum value is half of best, maximum is twice of best.

Table 3. Probability of a land cover type being classified as cropland or pasture, calculated using the

905 producer's accuracy, which is how often features on the ground are classified, or the probability that906 a certain pixel is classified as a given land cover class. Probabilities of being cropland or pasture cover

type ( $C_{cp}$ ) are adjusted based on patch size (A) for patches with A < 1 km<sup>2</sup>, where  $p(C_{cp}) = C_{cp} * A_{cp}^2$ .

| Value | Name                                 | Crop/ pastureland<br>weight | Proability<br>crop/pastureland |
|-------|--------------------------------------|-----------------------------|--------------------------------|
| 10    | Cropland, rainfed                    | 1                           | 0.887                          |
| 20    | Cropland, irrigated                  | 1                           | 0.893                          |
| 30    | Mosaic cropland (>50%)               | 0.5                         | 0.387                          |
| 40    | Mosaic cropland (>50%)               | 0.25                        | 0.366                          |
| 50    | Tree (>15%), broadleaved, evergreen  | 0                           | 0.038                          |
| 60    | Tree (>15%), broadleaved, deciduous  | 0                           | 0.070                          |
| 70    | Tree (>15%), needleleaved, evergreen | 0                           | 0.016                          |
| 80    | Tree (>15%, neeleeaved, deciduous    | 0                           | 0.000                          |
| 90    | Tree, mixed leaf type                | 0                           | 0.000                          |
| 100   | Mosaic tree/shrub (>50%)             | 0                           | 0.345                          |
| 110   | Mosaic herbaceous (>50%)             | 0                           | 0.091                          |
| 120   | Shrubland                            | 0                           | 0.104                          |
| 130   | Grassland                            | 0                           | 0.176                          |
| 140   | Lichens and mosses                   | 0                           | 0.000                          |
| 150   | Sparse vegetation (<15%)             | 0                           | 0.032                          |
| 160   | Tree, flooded                        | 0                           | 0.043                          |
| 170   | Tree, flooded saline                 | 0                           | 0.000                          |
| 180   | Shrub/herbaceous flooded             | 0                           | 0.000                          |
| 190   | Urban areas                          | 0                           | 0.120                          |
| 200   | Bare                                 | 0                           | 0.011                          |
| 210   | Water                                | 0                           | 0.018                          |
| 220   | Permanent snow & ice                 | 0                           | 0.000                          |

- Table 4. Summary of estimates of the degree of human modification (H) and the mean annualized
- difference between 5- or 10-yr increments for which change over time can be calculated (1990, 2000,
- 2010, and 2015), and H values for the contemporary dataset (~2017, all stressors). Mean annualized
- mean difference is calculated as the mean value across the continents of the difference in *H* values
- divided by the number of years (e.g.,  $H_{mad} = [H_{2015} H_{1990}]/25$ ).

|               | Mean H        |               |               |               | Mear          | ean annualized difference |                |                |               | ~2017         |               |  |
|---------------|---------------|---------------|---------------|---------------|---------------|---------------------------|----------------|----------------|---------------|---------------|---------------|--|
| Continent     | 1990          | 2000          | 2010          | 2015          | 1990-<br>2000 | 2000-<br>2010             | 2010-<br>2015  | 1990-<br>2015  | Med-<br>ian   | Mean          | Std.<br>Dev.  |  |
| Africa        | 0.0457        | 0.0489        | 0.0515        | 0.0530        | 0.00032       | 0.00026                   | 0.00030        | 0.00029        | 0.0056        | 0.1073        | 0.1730        |  |
| Asia          | 0.0856        | 0.0915        | 0.0988        | 0.1025        | 0.00059       | 0.00073                   | 0.00075        | 0.00067        | 0.0056        | 0.1542        | 0.2286        |  |
| Australia     | 0.0313        | 0.0324        | 0.0334        | 0.0341        | 0.00011       | 0.00011                   | 0.00013        | 0.00011        | 0.0006        | 0.0495        | 0.1250        |  |
| Europe        | 0.1145        | 0.1187        | 0.1206        | 0.1226        | 0.00042       | 0.00019                   | 0.00041        | 0.00033        | 0.0136        | 0.1533        | 0.2279        |  |
| No. America   | 0.0408        | 0.0419        | 0.0461        | 0.0463        | 0.00011       | 0.00042                   | 0.00005        | 0.00022        | 0.1309        | 0.1680        | 0.1681        |  |
| Oceania       | 0.0431        | 0.0475        | 0.0580        | 0.0662        | 0.00044       | 0.00105                   | 0.00164        | 0.00093        | 0.0527        | 0.1592        | 0.1856        |  |
| So. America   | 0.2378        | 0.2398        | 0.2434        | 0.2442        | 0.00020       | 0.00036                   | 0.00015        | 0.00026        | 0.2324        | 0.2868        | 0.2717        |  |
| <u>Global</u> | <u>0.0822</u> | <u>0.0864</u> | <u>0.0915</u> | <u>0.0946</u> | 0.00042       | <u>0.00051</u>            | <u>0.00062</u> | <u>0.00049</u> | <u>0.0096</u> | <u>0.1461</u> | <u>0.2146</u> |  |

- Table 5. A comparison of the mean annualized difference of human modification values for changes
- from 1990 to 2015 (H, 1990-2015), human footprint (HF, 1993-2009; Venter et al. 2016), and the
- temporal human pressure index (THPI, 1995-2010, Geldmann et al. 2019). Mean annualized mean
- difference is calculated as the mean value of the difference in *H* values divided by the number of
- 923 years (e.g.,  $H_{mad} = [H_{2015} H_{1990}]/25$ ), for each continent. Note that Oceania extends below Papau New
- Guinea (excluding the country of Australia).

| Continent     | НМ     | HF      | THPI    |
|---------------|--------|---------|---------|
| Africa        | 0.0003 | 0.0007  | 0.0011  |
| Asia          | 0.0007 | 0.0008  | 0.0012  |
| Australia     | 0.0001 | 0.0002  | 0.0001  |
| Europe        | 0.0003 | -0.0002 | 0.0002  |
| North America | 0.0002 | 0.0027  | -0.0001 |
| Oceania       | 0.0009 | 0.0011  | 0.0007  |
| South America | 0.0003 | 0.0000  | 0.0002  |
| <u>Global</u> | 0.0005 | 0.0006  | 0.0008  |

Table 6. A summary of the data, methods, and results comparing the degree of human modification

(HM; this paper); degree of human modification 1 km (HM1k; Kennedy et al. 2019a, 2019b, 2019c);

human footprint (HF; Sanderson et al. 2002; Venter et al. 2016); and temporal human pressure index

(THPI; Geldmann et al. 2019). Also see discussion of comparison in Kennedy et al. (2019b, 2019c),

Venter et al. (2019), and Riggio et al. (2020). Data source acronyms are provided in Table 1.

| Factor                                                 | НМ                                                                                                                                                                                                              | HM1k                                                                                                                                                                                               | HF                                                                                                               | ТНРІ                                                             |  |
|--------------------------------------------------------|-----------------------------------------------------------------------------------------------------------------------------------------------------------------------------------------------------------------|----------------------------------------------------------------------------------------------------------------------------------------------------------------------------------------------------|------------------------------------------------------------------------------------------------------------------|------------------------------------------------------------------|--|
| Conceptual<br>framework                                | Direct Threats Classification v2<br>(Salafsky et al. 2008)<br>Intensity values based on Land<br>Development Index (LDI; Brown<br>and Vivas 2005)                                                                | Direct Threats Classification<br>v2 (Salafsky et al. 2008)<br>LDI                                                                                                                                  | Sanderson et al.<br>(2002)                                                                                       | Geldmann et al.<br>(2014)                                        |  |
| Stressor:<br>Urban and<br>built-up                     | Urban and built-up (GHSL;<br>0.03-0.3 km; 1990-2015)                                                                                                                                                            | Urban and built-up (GHSL;<br>0.03-0.3 km; 2015)<br>Population density (GPW v4<br>2015, 1 km)                                                                                                       | Night-time lights<br>(DMSP/OLS >20; 1 km;<br>1994-2012)<br>Population density<br>CIESIN v3; 4 km; 1990,<br>2010) | Change in<br>population density<br>(GPW v3 1995,<br>2010, 1 km)  |  |
| Stressor:<br>Agriculture                               | Cropland & pastureland for 1990,<br>2015 (ESA CCI; 300 m) and<br>Cropland intensity (GLS, 1 km)<br>Unified Cropland Layer (UCL, 1<br>km)<br>Grazing (GLW, 10 km, 1 < livestock<br>units/km <sup>2</sup> < 1000) | Unified Cropland Layer<br>(UCL, 1 km)<br>Grazing (GLW v2, 1 km,<br>livestock units/km² < 1000)                                                                                                     | Cropland (UMD for<br>1990 and GlobCover<br>for 2009);<br>Pastureland (2000),<br>10 km                            | Cropland area<br>(HYDE, 10 km)                                   |  |
| Stressor:<br>Energy<br>production &<br>mining          | Oil & gas production (Gas flares<br>DMSP/OLS and VIIRS)<br>Renewable and non-renewable<br>power plants (WRI)<br>Large mining operations (S&P)                                                                   | Oil & gas wells, wind<br>turbines, mines (OSM, 2016,<br>VMAP0-2000)                                                                                                                                | N/A                                                                                                              | N/A                                                              |  |
| Stressor:<br>Transportatio<br>n & service<br>corridors | Road (highway, minor, two-track;<br>OSM, 2019)<br>Railways (OSM, 2019) Powerlines<br>(OSM, 2019)<br>Electrical power infrastructure<br>(harmonized DMSP/ VIIRS,<br>1992-2018)                                   | Road (highway, minor,<br>two-track, OSM, 2016,<br>gROADS-2000)<br>Railways (OSM, 2016,<br>VMAP0-2000), Powerlines<br>(OSM 2016)<br>Electric infrastructure<br>(night-time lights<br>DMSP-OLS-2013) | Roads (gROADS,<br>1980-2000); Railways<br>(VMAP-2000)<br>Electric infrastructure                                 | Nightlights<br>(DMSP/OLS<br>nightlights >20; 1<br>km; 1994-2012) |  |
| Stressor:<br>Biological<br>harvesting                  | Forest loss (Hansen, Curtis; 0.03-1<br>km, 2000-2017)                                                                                                                                                           | N/A                                                                                                                                                                                                | N/A                                                                                                              | N/A                                                              |  |
| Stressor:<br>Human<br>intrusions                       | Human intrusion (HUE, 1990-2015,<br>1 km)                                                                                                                                                                       | N/A                                                                                                                                                                                                | N/A                                                                                                              | N/A                                                              |  |
| Stressor:                                              | Reservoirs (GRanD, 1990-2017,                                                                                                                                                                                   | N/A                                                                                                                                                                                                | N/A                                                                                                              | N/A                                                              |  |

| Natural<br>system<br>modifications        | 0.03 km)                                                                                                             |                                                                                                                                     |                                                                                                 |                                         |
|-------------------------------------------|----------------------------------------------------------------------------------------------------------------------|-------------------------------------------------------------------------------------------------------------------------------------|-------------------------------------------------------------------------------------------------|-----------------------------------------|
| Stressor:<br>Pollution                    | Nitrous oxide pollution (EDGAR,<br>1990-2012, 100 km)                                                                | N/A                                                                                                                                 | N/A                                                                                             | N/A                                     |
| Metric                                    | Degree of human modification (H,<br>0-1.0 continuous value)                                                          | Degree of human<br>modification (H, 0-1.0<br>continuous value)                                                                      | Scaled 0-10, 0-4,<br>summed to 50,<br>ordinal value                                             | N/A                                     |
| Combine<br>factors                        | Increasive to 1.0 using fuzzy sum                                                                                    | Increasive to 1.0 using fuzzy sum                                                                                                   | Additive,<br>max-normalized                                                                     | Equal-weight,<br>additive<br>normalized |
| Uncertainty<br>or sensitivity<br>analysis | Calculates per-pixel variance due<br>to estimates of intensity values,<br>randomized (n=50)                          | Calculates per-pixel variance<br>due to estimates of<br>intensity values,<br>randomized (n=100)                                     | Sensitify of static v.<br>dynamic pasture data                                                  | N/A                                     |
| Validation                                | Tested using independent<br>validation dataset that included<br>~10,000 subplots within ~1,000 1<br>km² sample plots | Tested using independent<br>validation dataset that<br>included ~10,000 subplots<br>within ~1,000 1 km <sup>2</sup> sample<br>plots | Tested using<br>independent<br>validation dataset in<br>3,460 1 km <sup>2</sup> sample<br>plots | N/A                                     |

Table 7. Summary of results by biome, comparing trends using the mean annualized difference for the human modification ( $H_{mad}$ ), human footprint ( $HF_{mad}$ , Venter et al. 2016), and the mean temporal human pressure index ( $THPI_{mad}$ , Geldmann et al. 2019) score. Also provided are estimates of the proportion of terrestrial lands modified as estimated from Kennedy et al. (H1k; 2019), and HF (score was max-normalized to rescale to 0-1). The THPI dataset characterizes only change and so estimates of the proportion of lands modified in 2010 could not be provided. Mean annualized mean difference is calculated as the mean value across the continents and globally of the difference in H values

divided by the number of years.

| Biome name                                                  | HM<br>(1990-2015) | HF<br>(1993-2009) | THPI*<br>(1995-2010) | HM<br>(~2017) | HM**<br>(~2016) | HF<br>(2009) |
|-------------------------------------------------------------|-------------------|-------------------|----------------------|---------------|-----------------|--------------|
| Boreal Forests/Taiga                                        | 0.00000           | -0.00001          | 0.00000              | 0.0213        | 0.0374          | 0.0288       |
| Deserts & Xeric Shrublands                                  | 0.00001           | 0.00003           | 0.00003              | 0.0571        | 0.1059          | 0.0820       |
| Flooded Grasslands & Savannas                               | 0.00002           | 0.00002           | 0.00015              | 0.2024        | 0.2480          | 0.1423       |
| Mangroves                                                   | 0.00005           | 0.00005           | 0.00002              | 0.2165        | 0.3051          | 0.1972       |
| Mediterranean Forests, Woodlands &<br>Scrub                 | 0.00003           | 0.00008           | 0.00012              | 0.2903        | 0.3373          | 0.2162       |
| Montane Grasslands & Shrublands                             | 0.00001           | 0.00006           | 0.00006              | 0.0894        | 0.1634          | 0.1076       |
| Temperate Broadleaf & Mixed Forests                         | 0.00002           | 0.00003           | 0.00002              | 0.3744        | 0.3968          | 0.2485       |
| Temperate Conifer Forests                                   | 0.00002           | 0.00001           | 0.00006              | 0.1072        | 0.1561          | 0.0992       |
| Temperate Grasslands, Savannas &<br>Shrublands              | 0.00002           | 0.00001           | 0.00009              | 0.2374        | 0.2943          | 0.1668       |
| Tropical & Subtropical Coniferous<br>Forests                | 0.00003           | 0.00000           | 0.00025              | 0.2052        | 0.2606          | 0.1568       |
| Tropical & Subtropical Dry Broadleaf<br>Forests             | 0.00005           | 0.00012           | 0.00006              | 0.3317        | 0.4242          | 0.2265       |
| Tropical & Subtropical Grasslands,<br>Savannas & Shrublands | 0.00002           | 0.00006           | 0.00008              | 0.1476        | 0.2120          | 0.1207       |
| Tropical & Subtropical Moist<br>Broadleaf Forests           | 0.00005           | 0.00007           | 0.00009              | 0.1862        | 0.2310          | 0.1390       |
| Tundra                                                      | 0.00000           | 0.00000           | 0.00000              | 0.0023        | 0.0001          | 0.0066       |

# 942 Figure captions

- Figure 1. A comparison of the recent trends in human activities by ecoregion using the mean
- annualized difference estimated by: (a) human modification (H, from 1990-2015); (b) human footprint
- (for 1993-2009, Venter et al. 2016); and (c) temporal human pressure index (for ~1995-2010,
- Geldmann et al. 2019). Note: interactive maps are available at:
- https://davidtheobald8.users.earthengine.app/view/global-human-modification-change.