# Peer review of "Earth transformed: detailed mapping of global human modification from 1990 to"

_Earth System Science Data, 2019_

## Referee Comment (RC1) · Anonymous Referee #1 · 24 Apr 2020

Authors claim to "… provide the most granular, contemporary, comprehensive, high-quality, and robust data currently available to assess temporal and spatial trends of global human modification." This reviewer finds no reason to doubt their motivation, few reasons to doubt skill and tools applied, but only small evidence that authors achieved their goal. Without substantial changes and improvements one suspects they will miss that goal. They include a useful and thoughtful 'Caveats' section (Section 4.2) but without sufficient information to allow this reviewer to determine whether caveats outweigh results.

Data access: Dryad requires too much time to deliver 4 GB. Could the authors provide some kind of teaser data product, demonstrating their tools and techniques, that does not require reviewer / user to commit to 4 GB? Something labelled and protected under a separate DOI, accessible via a trusted competent repository? Perhaps the 2017 data? Or a monthly portion of it? As I work through this review, Dryad has not delivered notice of availability for this data product for more than nine hours. By any standards, this represents failure to meet my, much less ESSD's, expectations for data accessibility.

I worry very much about source data. Authors have provided a list in Table 1, but the more-useful reference list - with DOI! - occurs in Data Availability Section 4.3. Authors (page 12, section 2.4) refer to use of GE Engine. Because I know that several of their data sources did not (do not?) exist on GE Engine, I conclude that these authors uploaded some of their required products into GE Engine in order to then use GE Engine tools for analysis and display. Note reference to GE Engine version of Figures 1 and 3! Knowing Google, we have no confidence that GE Engine will have same version of same data products or the same tools available for a subsequent user? Section 4.3 implies reliability, while GE Engine - for all its strengths - implies proprietary changes? In other cases, I believe ESSD has asked authors to archive a snapshot in a repository separate from GE Engine to ensure a stable version matched to their description. Here we would at least need to know which and how many of the data products listed in Section 4.3 remain accessible via GE Engine or, more likely, whether subsequent users would need to track and obtain individual sources to replicate this work. From ESSD guidelines (https://doi.org/10.5194/essd-10-2275-2018) one reads "The journal expects that a future user, 5 or more years after date of publication, will find exactly the data, the tools and the recipe (description) that allow her or him to completely and reliably reproduce any figure from the original data description or accompanying research paper." These authors fail to convince us that they have met that ESSD expectation?

Manuscript fails to present comprehensive estimates of uncertainty. Throughout a reader encounters percent changes to four sig figs (e.g. 15.04% to take but one example of dozens), apparently precise to 0.01%. But, if these overall estimates instead carry an uncertainty larger than 0.01% (likely!), then single estimates of 15.04% or time changes of 0.60% need plus/minus uncertainty estimates? This reviewer doubts, for example, that authors can reliably report time-dependent changes from e.g. 2000 to 2015 for most regions. Instead, convert all data to area numbers ($km^2$) and then provide uncertainties in $\pm$ $km^2$? In some cases authors seem to calculate percent changes of percent data? At page 13 line 452, authors claim _with no basis or justification_ "estimate of the level of precision (~0.00001) [for H] given the data inputs". Where does this unlikely precision come from? Later still, page 14 line 494, the authors claim "our best-estimate of 0.14605: [for H]. Please explain? The first based purely on high n count of H estimates while the second based on 50 randomized map (all pixels?) comparisons? Authors provide no basis for confidence, nor any discussion of larger uncertainties due to other complicating factors (e.g. wildfires, climate warming, etc.). Present Section 3.4 focuses entirely and only on statistical uncertainties of H but misses larger questions of reliability and accuracy. Neither do the authors assign any uncertainty to so-called validation products HF or THPI. This reader gains no confidence from those comparisons; improvements such as exist occur entirely due to higher spatial resolution (e.g. page 14 line 504, 505)? Manuscript needs to

provide readers and users an expanded detailed section conveying accurate dissection of uncertainties in H - including cumulative uncertainties propagated forward from source data - and of overall uncertainties of change estimates with reference to factors (stressors) not included here. For calendar year 2019, combination of fires in Australia, Amazonia, Siberia, California, etc. represented an equal or larger stressor than any itemized here? True, or not true? Small areas with high intensity values? Cumulative impact over 15 years? Authors give no hint. Caveats (Section 4.2) addresses these issues but in qualitative narrative rather than quantitative form. I especially worry about this statement (page 16 line 550, 551: "estimates of H generated for areas less than roughly 100 km 2 should be used with caution"! How then does a reader have any confidence in 0.09 km² resolution?

Specific comments, suggestions, complaints

Page 2 lines 23-24, "natural lands were lost (~17 per minute)" 17 pitches per minute? At 8 pitches per breath that means only 2 very slow breathes per minute? Back of the envelope: 5 slow breathes per minute, 7k m² per pitch, gives 5 x 7k x 8 = 280k m², = 0.28 km² per minute? Times 1440 minutes per day, would give 400 km² per day. If I work backward from their number - 178 km² per day - that gives .12 km2 per minute, or - at 7k m² per pitch and 8 pitches per breath, closer to 2 breathes per minute? Can the authors sustain fewer than 6 breathes per minute? Sorry to waste my and the authors time on this triviality, but unless they declare units more clearly and accurately, this sentence represents a not-useful diversion? Unfortunately, despite enthusiasm expressed in these 'real world' areal estimates, uncertainty here early in the manuscript persists throughout.

Page 3 line 49: Interestingly, the HYDE 3.2 product (by same author - Klein Goldewijk - cited here, also in ESSD at https://doi.org/10.5194/essd-9-927-2017), attempts similar assessments of total land use impacts from an inventory approach different to the remote sensing approach applied here. That group aspires to much longer (10k years) time coverage at much lower spatial resolution but justifies the trade-off of low spatial resolution for extended temporal coverage. Not 'high-resolution" but potentially "temporally comparable" at least over past decades? Have these authors have dismissed too many prior studies or contemporary work on human impacts issues?

Page 3 line 60: "obstructions by vegetation canopy (e.g., some roads, trails)" - Strange wording here. I think the authors mean that remote sensing might miss development features such as roads or trails hidden by vegetation canopies? As written, however, the sentence implies that roads and trails represent examples of vegetation canopies?

Page 3 line 69: "assumes additive but monotonic relationships" additive but monotonic? Some inconsistency here? Authors need to provide better explanation?

Page 8 line 264: "silver, tin, U 3 O 8 , and zinc" - strange to find uranium represented by chemical formula of uranium oxide while authors list all other metals by common names?

Page 10 line 337: I understand why authors felt the need to not include wildfires, but this exclusion should show up as an uncertainty later? (ESSD published a summary of global fire emissions, which must have had some satellite-based global fire product as it basis?)

Page 12 line 418: here 177 while the abstract had 178 (if "roughly", use 175?) and again the 17 football pitches not useful.

Page 12 lines 423, 424: absence of climate-induced warming as one of their stressors becomes acute for high latitude regions: Arctic greening (one of the predominant satellite-detected changes on the planet even if now waning); permafrost thaw; loss of snow cover

(global gridded product does exist going back two or three decades)? At lower latitudes, sea-level driven salt water intrusion of mangroves or into island freshwater aquifers, etc? In Oceania, large scale consumption-driven conversion to palm oil plantations (I think I saw a recent ESSD data product on this), possibly a more-useful validation point? Invasive species, food security, social health likewise, even if global data for those features does not exist? Despite skill with selected data products, authors have not convinced this reader that they captured the most important stressors. Despite basic oft-repeated excuse - that global data sets of not-included stressors do not exist - they have not convinced us that compilation and analysis of global data products that do exist represents a reliable reproducible useful product

---

## Referee Comment (RC2) · Anonymous Referee #2 · 4 May 2020

This new dataset represents an important advancement in our assessment of natural areas and human impacts. It will likely be useful to numerous global evaluations of conservation priority, threats, modeling potential processes, and structural connectivity (among others). Taking inventory of how humans impact nature is among the most important endeavors guiding how humans steward the planet. Thank you for this contribution.

My only major concern with the paper is how water is dealt with. If flow is regulated the water surface gets a human modification value, but if the water is unregulated the water surface gets an NA. This results in some odd patterns. For instance one of the Great Lakes is filled with values, but the other four are empty. Yellowstone Lake is empty but Jackson Lake in Grand Teton National Park, which is partially regulated
Interactive
comment

has values. One of the key uses of these kinds of data products is to model connectivity of ecological processes or species movement. High human modification is associated with behavioral avoidance or higher risk of mortality. These kids of gridded data are used frequently for these purposes. I recommend that the authors at least produce a version that has all water bodies masked out. I just read this paper that showed that dealing with water bodies is important for connectivity modeling: https://www.nature.com/articles/s41598-020-63545-z

One small point. I found the sentence in the abstract about "over the pause of a deep breath" confusing. I had to read it several times to understand what they meant. I think they can make the same point - which is a good one - with different wording.

---

## Author Comment (AC1) · 6 Jun 2020

June 5, 2020

**Responses to Reviewer #1**

Thank you for your review of our paper: ESSD-2019-252. Below we paraphrase your comments in bold and provide our responses in regular text. We also recognize the challenging context of the COVID-19 pandemic as well, and are most appreciative of your review.

**Needs substantial revision before acceptable to ESSD.**
**Data access comments, related to difficulty downloading the data from the Dryad repository, including suggestions to provide a "teaser data product" and a more prompt delivery of data.**
We became aware of these issues when a few scientists contacted us about the data repository. We found that Dryad does in fact respond promptly, but because of the large file size sends an email with a URL for users to download the data. We have discovered that this email, unfortunately, is frequently filtered into the Spam folder. We added a note in the data description portion of the Dryad repository material to alert users to this situation. We note that the Dryad repository meets the requirements specified by ESSD. Part of the challenge is simply due to the fine-resolution global datasets and floating-point values that do not compress well. Also, while we recognize it isn't a permanent solution, we provided a URL:
([https://davidtheobald8.users.earthengine.app/view/global-human-modification-change](https://davidtheobald8.users.earthengine.app/view/global-human-modification-change)) for a dynamic mapping website that allows rapid visualization of our data, and comparison to a few other commonly referenced datasets.

**I am worried about source data availability, particularly in relation to the use of Google Earth Engine.**
You are correct that we did implement our analysis in Google Earth Engine (GEE) and did upload source data into the GEE platform to conduct our analysis. But, all source data used are open source and are accessible externally via the permanent DOIs that we provided. Furthermore, the formulas are carefully and clearly described, following the guidelines of ESSD in providing DOI permanent links to all source data and "recipe" used in the analysis to create the data product presented in the paper.
[Note to Editor: please provide guidance on this issue, as well as advising on what arrangement within the paper is preferred -- during the creation of the Discussion Paper we originally had a list of all datasets in the Data Availability section, but moved the citations with DOI into the References section in response to a request to streamline and provide acronyms in Table 1. We are open to either structure in the paper.]

**Manuscript fails to present comprehensive estimates of uncertainty.**
Thank you for your detailed comments on this important issue. In our response we address four aspects from the issue you raised regarding "uncertainty": (1) our ability to capture dynamic events, such as wildfire or climate change; (2) understanding the uncertainty of our results related to

measurement error; (3) the precision with which we report results; and (4) including uncertainty in our validation analysis. We address each of these in order:

1.  Uncertainties associated with dynamic vents, land uses, and activities that we did not attempt to capture. We address this briefly in the caveat section and by citing our previous work where we discuss these challenges, particularly around wildlife and climate change. We revised our text in the caveat section to read, on lines 560-570:

    > *"As with any model, we recognize there are limitations of our work. We did not include data for all human stressors, largely because of incomplete global coverage or coarse mapping units (Klein Goldewijk et al. 2007; Geldmann et al., 2014), an inability to discern human-induced versus natural disturbances, or uncertainty in the location and directionality of its impact (e.g.; climate change on terrestrial systems; Geldmann et al., 2014). In particular and discussed in Kennedy et al. (2019a, 2019b), changes to land cover due to ecological disturbance events, such as wildfires or flooding, are not included in our analysis because of the difficulty in separating natural from human-caused disturbances -- yet, we recognize that because of the broad extent of wildfire, in particular, would have strong implications. We did not include climate data as a stressor in this product to keep our analysis manageable and tractable. For more integrated analyses, our data product should be used in combination with datasets of impacts due to climate change (e.g., Parks et al. 2020)."*

2.  We have revised our manuscript to improve how we address how uncertainty affects our results for 2017 by conducting an additional analysis of the per-pixel variability (standard deviation) and adding summary results of it along with a map (in Figure 4), providing values across the randomized iterations of the mode. We also addressed this further in the text by reiterating key aspects of methodology in the revised Uncertainty and validation analysis section (quoted below in italics), in particular our directly including uncertainty into the formulas we used to calculate the human modification for each stressor. For convenience, we quote changes from our manuscript: To explain the context for our responses, here we briefly incorporate uncertainty for a number of key stressors: (a) using the results from the accuracy assessment of the land cover dataset, we adjusted weights associated with land cover types when estimating the degree of human modification (H); (b) similarly, we weighted our estimates for the urban/built-up stressor when calculating H, which was calculated as a function of the degree of confidence of the modeled estimates as provided by the GHSL dataset, on a per-pixel basis; (c) we addressed the spatial uncertainty associated with stressors represented as points (e.g., mine locations, gas flares) and lines (i.e. roads) when calculating H. We added Figure 4 and revised our text in the caveat section to read, on lines 502-512:

    > *"To address uncertainty in our results, we specifically included an estimate of uncertainty associated with each stressor in the calculations of human modification for 2017 conditions (Equations 4-26). For example, we adjusted $p(C_{cp})$ by directly incorporating measured confusion among land cover types using the results from the accuracy assessment of the land cover dataset (from Eq. 4).*

*We summarized the uncertainty maps by calculating the global mean for each of the 50 randomizations, and found the mean of the 50 global mean values was 0.1434 (SD= ±0.0076) and ranged from 0.1243 to 0.1612, thus, in line with the global mean of 0.1461 obtained using our "best-estimate" intensity values. We also mapped the per-pixel variance (standard deviation) to examine the spatial pattern of uncertainty (Figure 4). The locations of the highest levels of uncertainty tend to be in more highly developed landscapes."*

3. We responded to your comment about our reporting of results without providing variance measures and overly-high precision by modifying our text to provide our results of our measure of human modification (which ranges from 0.0 to 1.0) to be consistent with reporting percentages using 4 orders of precision (i.e., +/- 0.0001 rather than +/- 0.00001). As suggested, we also added a +/- when reporting our estimates of human modification in terms of area (i.e. square kilometers). This includes removing our statement that you found troubling regarding the 100 $km^2$ unit of analysis area. These changes occur on lines 28, 422, 435, 450-462, 508-9, and Tables 4-6.

4. We addressed your comment about validation: "Neither do the authors assign any uncertainty to so-called validation products HF or THPI", by clarifying the purpose of our validation analysis and how we accomplished it. We revised our manuscript text to clarify our steps, and included further citations that provide additional details. To be clear, we did not validate our results against the modeled outputs from the human footprint (HF) or human pressure index (THPI), rather, we simply *compared* our data to them because they are typically perceived as being similar, are readily-available and frequently used datasets, in anticipation that this will be a common and reasonable question of readers and data users. In fact, a central reason we have produced the work in this manuscript is to build on and provide a more refined, and we believe and argue here, improved way to spatially represent and measure the degree of human modification on landscapes. By calculating and reporting the coefficient of determination (i.e. $r^2$), we quantified how well our "ground truth" data, described in Kennedy et al. (2019a, 2019b) were replicated by our estimates of H for 2015. In Kennedy et al. (2019c), we conducted additional statistical analysis and more fully described our methods of further examination of the resulting distributions. To this end, we made changes in lines 514-520:.

*"We found strong agreement between H for ~2017 and our validation data (r=0.783), with an average root-mean-square-error of 0.22 and a mean-absolute-error of 0.04, for the 926 ~1 $km^2$ plots (9,260 sub-plots). There were 726 plots within ±20% agreement, while for 161 plots H was estimated higher than our visual estimate from the validation data (and 39 plots lower). Estimates of H were biased high, likely because the stressors for the "human intrusion" and electrical infrastructure (based on nighttime lights) are not readily observable from the aerial imagery used to generate the validation data. Our results here are consistent with our earlier findings (Kennedy et al. 2019a, 2019b, 2019c)."*

**Specific comments and suggestions below:**

**Page 2, lines 23-24, question about the duration of a breath:**
Thank you -- following your questions and Reviewer #2's suggestion, we removed this "real-world" comparison from the manuscript.

**Page 3, Line 49: Have [we] dismissed too many prior studies or contemporary work on human impact issues?**
Thank you -- at your suggestion, we cited a few additional important works in the field, in particular the work on HYDE 3.2 product ([https://doi.org/10.5194/essd-9-927-2017](https://doi.org/10.5194/essd-9-927-2017)), Ellis' work on mapping the Anthromes, and recent work (Riggio et al. 2020) that compares human modification (Kennedy et al. 2019a), Human Footprint, Anthromes, as well as Jacobsen's (Jacobsen et al. 2019) and Riggio's work (Riggio et al. 2020). It is always a challenge to balance providing enough context for when developing new science. We chose to provide a more focused, technical description as the purpose of our paper is to develop a specific data resource that examines recent change (1990-2015) and relatively high-resolution for global work (0.3 km) -- rather than a broader review of similar previous efforts.

**Page 3, line 60: Clarify wording: "...obstructions by vegetation canopy (e.g., some roads, trails)".**
Good suggestion, we modified lines 59-62 to read:
> *"This is because remotely sensed imagery has limitations for this application -- especially prior to ~2010 -- because it can require human-interpretation to classify adequately and can miss development features that are obstructed by vegetation canopy or  are small or narrow features (e.g., towers, wind turbines, powerlines)."*

**Page 3, Line 69: clarify explanation of "...additive but monotonic relationships..."**
Thanks, we simplified this sentence to clarify it on line 70, to read: "...measure that assumes additive relationships among stressors..."

**Page 8, line 264: be consistent when listing metals with common names.**
Thanks, we modified our text on line 266 to state "uranium oxide".

**Page 10, line 337: Clarify why wildfires, if excluded in the analysis, do not show up as an uncertainty.**
Good point, we added text to describe how wildfire (and other dynamic ecological processes) are considered within our work on lines 339-342:
> *"(Note that we excluded wildfire as a stressor because of the challenges of attributing wildfires to human causation-- especially over global extent, and urbanization because it is measured directly by the built-up stressor)."*

and lines 564-570:
> *"In particular and discussed in Kennedy et al. (2019a, 2019b), changes to land cover due to ecological disturbance events, such as wildfires or flooding, are not included in our analysis because of the difficulty in separating natural from human-caused disturbances -- yet, we recognize that because of the broad extent of wildfire, in particular, would have strong*

*implications. We did not include climate data as a stressor in this product to keep our analysis manageable and tractable. For more integrated analyses, our data product should be used in combination with datasets of impacts due to climate change (e.g., Parks et al. 2020)."*

**Page 12, line 418: clarify and be consistent with area estimate, and remove reference to football pitches.**
Thanks, done.

**Page 12, lines 423-4. Clarify why available climate change datasets are not used.**
Thanks, this is an important point. We agree that climate change effects are happening, and there are numerous climate data products and a burgeoning field of science. To address this point, we clarified our decision not to include it in our analysis on lines 116-123:

> *"We note that we did not map stressors for invasive species or pathogens and genes, geologic events, or climate change. This was because suitable temporal global data were not available to capture stressors due to invasive species or pathogens and genes; the majority of geological events are not directly caused by humans; and climate change is better modeled as separate process distinct from the effects of direct human activities and has a plethora of research on this topic (Geldmann et al. 2014; Titeux et al. 2016)."*

and on lines 568-570:

> *"We did not include climate data as a stressor in this product to keep our analysis manageable and tractable. For more integrated analyses, our data product should be used in combination with datasets of impacts due to climate change (e.g., Parks et al. 2020)."*

Sincerely,

David M. Theobald, Ph.D., on behalf of co-authors

---

## Referee Comment (RC3) · Anonymous Referee #3 · 9 Jun 2020

**General comments**

Theobald et al., mapped the temporal change and the "current" state of the degree of human modification, H using the Direct Threats Classification v2 (Salafsky et al. 2008; cmp-openstandards.org) methodology. The presented dataset can be highly valuable for both research and decision making. However, some clarifications and revisions are needed beside the concerns already raised by Reviewer 1 and 2. Most importantly, I recommend the authors to share the complete dataset and clearly describe the dataset provided.

**Specific comments**
*Data*
It would be useful if the authors could provide a readme file (or improve the usage note

description on Dryad) for the data provided, perhaps a table listing what files and data are actually shared. (I only tried to check on the data using Python, not sure if use of e.g., Google Earth Engine would have showed up anything differently. If there are differences, the authors could perhaps try to bridge the differences or recommend a preferred software.)

It was for example unclear to me:

- What does each of the three data folders represent? E.g., what does 60c and 60s stand for in the folder names?

- Within the folders, there are several files. A readme file could describe what each of these files contain.

- (It could be something wrong on my side, but I could only plot the Oceania files with the ending "0000000000-0000032768", whereas the 0000000000-0000000000 files threw the following error: Readorwritefailed. gHMv1_1990_1000_60c_land-0000000000-0000000000.tif,band1: IReadBlockfailedatXoffset121,Yoffset39:TIFFReadEncodedTile()failed.).

- The manuscript also mentions a recent 2017 dataset, but none of the folder names contain "2017" (only 1990 and 2015).

- The manuscript states that "global datasets for 1990, 2000, 2010, and 2015" are provided, but it's unclear which files contain the 2000 and 2010 data?

- The embedded metadata could be complemented with lat, lon, date, and unit information?

- The data description mentions the change stressors and the uncertainty analyses, but are these really included in the data deposited at Dryad? It would be useful if the authors shared the individual stressors, for users who might want to inspect the importance and contribution of individual stressors to the overall H.

*Manuscript*

Perhaps consider adding a time index for the variables that vary in the stressor equations. For instance, in the Urban and Built-up (L182-) authors state that the probability on GHSL stressor p(C) is based on 2014 data, seem to suggest that years for Ibu varies according to Table 2, and do not specify the year for Fbu. Adding a time index (e.g., u or t, as in Eq 3) would help the reader see which variables are kept constant and which ones are varying according to Table 2.

As there are several comparisons made with the Human Footprint (HF) and the temporal human pressure index (THPI), it might be useful to also provide an overview table or systematic description of how these datasets differ or are similar to each other in terms of their methodology and data input use as well. This may facilitate interpretation of the comparison results.

P6L164: Hmed and Hmad are very similar. Perhaps consider chaing to Hmedian.

P13L433: "We found that about 19.1 Mkm2 of natural lands were lost by 2017". However, e.g., (Ramankutty, Evan, Monfreda, Foley, 2008) estimates that cropland occupies 15 million km2 and pasture 28 million km2), and FAO estimated that agricultural areas amount to about 50 million km2 in 2000 (http://www.fao.org/uploads/media/grass_stats_1.pdf). I would suggest the authors to comment on those differences, to not leave the reader wondering why the estimates differ so much.

P14L476: "we found that 14.5% of terrestrial lands globally have been modified, which is roughly similar to HF". (12.3% for 2009; Venter et al. 2016). About half of the Earth's terrestrial surface has been transformed by humans, e.g., according to (Hurtt et al., 2006) " 42–68% of the land surface was impacted by land-use activities". Perhaps some comments on such difference might be insightful for the reader as well.

Table 1: Could you also clarify, perhaps in this table, which stressors were used for the 1990-2015 and the 2017 datasets, respectively? (Perhaps by splitting the "year" column to "years used in 1990-2015 dataset" and "years used in the 2017 dataset"). It

would be nice with a clear overview.

Related to Reviewer 1's comment on the usability of the data: possibly, you could consider a 0.5 degree resolution dataset for users who do not need the higher resolution version, and only wishes to do some quick initial inspections, comparisons, or visualizations, or those in the world who are not so lucky to have quick and reliable internet connection or adequate computer power. This is not a recommendation in anyway, just an idea for increasing usability.

**References**

Hurtt, G. C., Frolking, S., Fearon, M. G., Moore, B., Shevliakova, E., Malyshev, S., . . . Houghton, R. a. (2006). The underpinnings of land-use history: three centuries of global gridded land-use transitions, wood-harvest activity, and resulting secondary lands. Global Change Biology, 12(7), 1208–1229. https://doi.org/10.1111/j.1365-2486.2006.01150.x

Ramankutty, N., Evan, A. T., Monfreda, C., Foley, J. A. (2008). Farming the planet: 1. Geographic distribution of global agricultural lands in the year 2000. Global Biogeochemical Cycles, 22(1), 1–19. https://doi.org/10.1029/2007GB002952

––––––––––––––––––––––––––

---

## Author Response (AR1)

June 25, 2020

**Responses to Reviewer #1**

Thank you for your review of our paper: ESSD-2019-252. Below we paraphrase your comments in bold and provide our responses in regular text. We also recognize the challenging context of the COVID-19 pandemic as well, and are most appreciative of your review.

**Needs substantial revision before acceptable to ESSD.**

Data access comments, related to difficulty downloading the data from the Dryad repository, including suggestions to provide a "teaser data product" and a more prompt delivery of data. We became aware of these issues when a few scientists contacted us about the data repository. We found that Dryad does in fact respond promptly, but because of the large file size sends an email with a URL for users to download the data. We have discovered that this email, unfortunately, is frequently filtered into the Spam folder. We added a note in the data description portion of the Dryad repository material to alert users to this situation. We note that the Dryad repository meets the requirements specified by ESSD. Part of the challenge is simply due to the fine-resolution global datasets and floating-point values that do not compress well. Also, while we recognize it isn't a permanent solution, we provided a URL:

(https://davidtheobald8.users.earthengine.app/view/global-human-modification-change) for a dynamic mapping website that allows rapid visualization of our data, and comparison to a few other commonly referenced datasets.

**I am worried about source data availability, particularly in relation to the use of Google Earth Engine.**

You are correct that we did implement our analysis in Google Earth Engine (GEE) and did upload source data into the GEE platform to conduct our analysis. But, all source data used are open source and are accessible externally via the permanent DOIs that we provided. Furthermore, the formulas are carefully and clearly described, following the guidelines of ESSD in providing DOI permanent links to all source data and "recipe" used in the analysis to create the data product presented in the paper. Please note that we placed citations with DOI into the References section to streamline and make a more concise document by providing acronyms in Table 1.

**Manuscript fails to present comprehensive estimates of uncertainty.**

Thank you for your detailed comments on this important issue. In our response we address four aspects from the issue you raised regarding "uncertainty": (1) our ability to capture dynamic events, such as wildfire or climate change; (2) understanding the uncertainty of our results related to measurement error; (3) the precision with which we report results; and (4) including uncertainty in our validation analysis. We address each of these in order:

1. Uncertainties associated with dynamic events, land uses, and activities that we did not attempt to capture. We address this briefly in the caveat section and by citing our previous

work where we discuss these challenges, particularly around wildlife and climate change. We revised our text in the caveat section to read, on lines 563-573:

"As with any model, we recognize there are limitations of our work. We did not include data for all human stressors, largely because of incomplete global coverage or coarse mapping units (Klein Goldewijk et al. 2007; Geldmann et al., 2014), an inability to discern human-induced versus natural disturbances, or uncertainty in the location and directionality of its impact (e.g.; climate change on terrestrial systems; Geldmann et al., 2014). In particular and discussed in Kennedy et al. (2019a, 2019b), changes to land cover due to ecological disturbance events, such as wildfires or flooding, are not included in our analysis because of the difficulty in separating natural from human-caused disturbances -- yet, we recognize that the broad extent of wildfire in particular, could have strong implications. We did not include climate data as a stressor in this product to keep our analysis manageable and tractable. For more integrated analyses, our data product should be used in combination with datasets of impacts due to climate change (e.g., Parks et al. 2020)."

2. We revised our manuscript to improve how we address how uncertainty affects our results for 2017 by conducting an additional analysis of the per-pixel variability (standard deviation) and adding Figure 4 which provides a map as well as summary results, providing values across the randomized iterations of the mode.

Additionally, we realize that a few of the uncertainty measures we incorporated were dispersed in the methodology section when describing the modeling approach. We therefore added text in the revised Uncertainty and validation analysis section (quoted below in italics) to reiterate key aspects of the methodology that directly include uncertainty in the formulas used to calculate the human modification for each stressor. In particular we: (a) used the results from the accuracy assessment of the land cover dataset, to adjust weights associated with land cover types when estimating the degree of human modification (H); (b) similarly, we weighted our estimates for the urban/built-up stressor when calculating H, as a function of the degree of confidence of the modeled estimates provided by the GHSL dataset, on a per-pixel basis; and (c) addressed the spatial uncertainty associated with stressors represented as points (e.g., mine locations, gas flares) and lines (i.e. roads) when calculating H.

The above mentioned revisions are included in Figure 4 and the revised text on lines 506-515: "We addressed uncertainty in our results by incorporating the parameter  $p(C_s)$  for every sector s to best quantify uncertainty in its spatial location and classification as detailed in section 2.2.; for example, we adjusted  $p(C_{cp})$  by directly incorporating measured confusion among land cover types using the results from the accuracy assessment of the land cover dataset (from Eq. 4). Additionally, we incorporated uncertainty by calculating the global mean for each of the 50 randomizations, which across the 50 iterations was 0.1434 (SD= ±0.0076) and ranged from 0.1243 to 0.1612. Thus, the global mean of 0.1461 obtained using our "best-estimate" intensity values was in line with our uncertainty results. We also mapped the per-pixel variance (standard deviation) to examine the spatial pattern of uncertainty (Figure 4). The locations of the highest levels of uncertainty tend to be in more highly developed landscapes."

- 3. We responded to your comment about our reporting of results without providing variance measures and overly-high precision by modifying our text to include a measure of human modification (which ranges from 0.0 to 1.0) using 4 orders of precision (i.e., +/- 0.0001 rather than +/- 0.00001) to be consistent with reporting percentages. As suggested, we also added a +/- when reporting our estimates of human modification in terms of area (i.e. square kilometers). This includes removing our statement that you found troubling regarding the 100 km2 unit of analysis area. Changes occur on lines 28, 422, 435, 451-463, 511-12, and Tables 4-6.
- 4. We addressed your comment about validation: "Neither do the authors assign any uncertainty to so-called validation products HF or THPI", by clarifying the purpose of our validation analysis and how we accomplished it. We revised our manuscript text to clarify our steps, and included further citations that provide additional details. To be clear, we did not validate our results against the modeled outputs from the human footprint (HF) or human pressure index (THPI), rather, we simply *compared* our data to them because they are typically perceived as being similar, are readily-available and frequently used datasets, and we anticipated such a comparison will be a common and reasonable question of readers and data users. In fact, a central reason we have produced the work in this manuscript is to build on and provide a more refined and improved way to spatially represent and measure the degree of human modification on landscapes. That said, while we compared our data to other available products, we note that we did indeed validate our data by calculating and reporting the coefficient of determination (i.e. *r*2) against "ground truth" data described in Kennedy et al. (2019a, 2019b) i.e., "our validation data" mentioned in lines 517-523:

"We found strong agreement between H for ~2017 and our validation data (r=0.783), with an average root-mean-square-error of 0.22 and a mean-absolute-error of 0.04, for the 926 ~1 km2 plots (9,260 sub-plots). There were 726 plots within  $\pm$ 20% agreement, while for 161 plots H was estimated higher than our visual estimate from the validation data (and 39 plots lower). Estimates of H were biased high, likely because the stressors for the "human intrusion" and electrical infrastructure (based on nighttime lights) are not readily observable from the aerial imagery used to generate the validation data. Our results here are consistent with our earlier findings (Kennedy et al. 2019a, 2019b, 2019c)."

**Specific comments and suggestions below:**

**Page 2, lines 23-24, question about the duration of a breath:**

Thank you -- following your questions and Reviewer #2's suggestion, we removed this "real-world" comparison from the manuscript.

**Page 3, Line 49: Have [we] dismissed too many prior studies or contemporary work on human impact issues?**

Thank you -- at your suggestion, we cited a few additional important works in the field, in particular the work on HYDE 3.2 product (https://doi.org/10.5194/essd-9-927-2017), Ellis' work on mapping the Anthromes, and recent work (Riggio et al. 2020) that compares human modification (Kennedy et al. 2019a), Human Footprint, Anthromes, as well as Jacobsen's (Jacobsen et al. 2019) and Riggio's work (Riggio et al. 2020). It is always a challenge to balance providing enough context for when developing new science. We chose to provide a more focused, technical description as the purpose of our paper is to develop a specific data resource that examines recent change (1990-2015) and relatively high-resolution for global work (0.3 km) -- rather than a broader review of similar previous efforts.

**Page 3, line 60: Clarify wording: "...obstructions by vegetation canopy (e.g., some roads, trails)".** Good suggestion, we modified lines 59-62 to read:**

"This is because remotely sensed imagery has limitations for this application -- especially prior to ~2010 -- because it can require human-interpretation to classify adequately and can miss development features that are obstructed by vegetation canopy or are small or narrow features (e.g., towers, wind turbines, powerlines)."

**Page 3, Line 69: clarify explanation of "...additive but monotonic relationships..."**

Thanks, we simplified this sentence to clarify it on line 70, to read: "...measure that assumes additive relationships among stressors..."

**Page 8, line 264: be consistent when listing metals with common names.**

Thanks, we modified our text on line 266 to state "uranium oxide".

**Page 10, line 337: Clarify why wildfires, if excluded in the analysis, do not show up as an uncertainty.** Good point, we added text to describe how wildfire (and other dynamic ecological processes) are considered within our work on lines 339-342:

"(Note that we excluded wildfire as a stressor because of the challenges of attributing wildfires to human causation-- especially over global extent, and urbanization because it is measured directly by the built-up stressor)."

and lines 567-573:

"In particular and discussed in Kennedy et al. (2019a, 2019b), changes to land cover due to ecological disturbance events, such as wildfires or flooding, are not included in our analysis because of the difficulty in separating natural from human-caused disturbances -- yet, we recognize that the broad extent of wildfire in particular, could have strong implications. We did not include climate data as a stressor in this product to keep our analysis manageable and tractable. For more integrated analyses, our data product should be used in combination with datasets of impacts due to climate change (e.g., Parks et al. 2020)."

**Page 12, line 418: clarify and be consistent with area estimate, and remove reference to football pitches.**

Thanks, done.

**Page 12, lines 423-4. Clarify why available climate change datasets are not used.**

Thanks, this is an important point. We agree that climate change effects are happening, and there are numerous climate data products and a burgeoning field of science. To address this point, we clarified our decision not to include it in our analysis on lines 116-123:

"To estimate the current amount of H circa 2017 year (median=2017, min=2012, max=2019), we included three additional stressors, including grazing, oil and gas wells, and powerlines. We note that we did not map stressors for invasive species or pathogens and genes, geologic events, or climate change. This was because suitable temporal global data were not available to capture stressors due to invasive species or pathogens and genes; the majority of geological events are not directly caused by humans; and climate change is better modeled as separate process distinct from the effects of direct human activities and has a plethora of research on this topic (Geldmann et al. 2014; Titeux et al. 2016)."

**and on lines 571-573:**

"We did not include climate data as a stressor in this product to keep our analysis manageable and tractable. For more integrated analyses, our data product should be used in combination with datasets of impacts due to climate change (e.g., Parks et al. 2020)."

Sincerely,

David M. Theobald, Ph.D., on behalf of co-authors

June 25, 2020

**Reviewer #2**

Thank you for your review of our paper: ESSD-2019-252. Below we paraphrase your comments in bold (for clarity), and provide our responses in regular text. We also recognize the challenging context of the COVID-19 pandemic as well, and are most appreciative of your review.

This new dataset represents an important advancement in our assessment of natural areas and human impacts... Thank you for this contribution. Thanks!

**My only concern is how water is dealt with, that for some uses such as modeling ecological processes or species movement it would be valuable to at least produce a version of the dataset that has all water bodies masked out.**

This is an important comment, and we appreciate your suggestion to provide a second version of the data with water bodies removed. We believe including reservoirs to include the modification of ecological processes through habitat loss and fragmentation of the artificial water bodies is an important advance. Yet, we recognize there can be situations where removing all waterbodies may be more suitable for a given problem. To address this, we also added a separate land/water mask dataset to the repository to allow users to readily remove reservoirs so as to treat all waterbodies in a similar fashion. We updated the manuscript to note this addition in the Data Availability section.

**A minor point -- revise the sentence containing: "...over the pause of a deep breath".**

Thank you -- we have revised to read more simply: "... over 12 hectares each minute."

Sincerely,

David M. Theobald, Ph.D., on behalf of co-authors

June 25, 2020

**Responses to Reviewer #3**

Thank you for your review of our paper: ESSD-2019-252. Below we paraphrase your comments in bold and provide our responses in regular text. We also recognize the challenging context of the COVID-19 pandemic, and are most appreciative of your review.

**General comments: The presented dataset can be highly valuable for both research and decision making. However, some clarifications and revisions are needed beside the concerns already raised by Reviewer 1 and 2. Most importantly, I recommend the authors to share the complete dataset and clearly describe the dataset provided.**

Thank you. To address your general comment, we have revised our dataset repository (DOI) to clarify and provide additional details on major stressor types to document file naming and structure. In addition, we respond to specific comments that are related to this general one below.

**Provide datasets for the individual stressors that comprise the overall human modification dataset**

We grouped individual stressors into major categories: urban/built-up, agriculture, energy/mining, and transportation/service, and all individual stressor data are readily available and/or are fully documented in our paper. Providing a copy of those data in our repository is somewhat redundant and unfortunately in some cases, would verge on license issues.

**Data repository: It would be useful if the authors could provide a readme file (or improve the usage note description on Dryad) for the data provided, perhaps a table listing what files and data are actually shared. (I only tried to check on the data using Python, not sure if use of e.g., Google Earth Engine would have shown anything differently. If there are differences, the authors could perhaps try to bridge the differences or recommend a preferred software.)**

Thank you for this suggestion. We have improved the usage note description associated with the Dryad repository (please refer to the revised text from that document above). We note here that there is no additional meta-data content provided in the Google Earth Engine view of these data.

**What do the folder names represent (e.g. "6oc vs. 6os")? What files are associated with each folder?**

We have simplified the datafile and folder structure, and described this structure fully in the data description of the repository, to the following. When unzipped, each zip file expands to a folder with the constituent files:

gHMv1\_300m\_1990\_change.zip gHMv1\_300m\_2000\_change.zip gHMv1\_300m\_2010\_change.zip gHMv1\_300m\_2015\_change.zip gHMv1\_300m\_2017\_static.zip gHMv1\_300m\_2017\_static\_stressors.zip

**gHM\_landLakeReservoirOcean300m.zip**

**Regarding potential error on file gHMv1\_1990\_1000\_60c\_land-000000000-000000000...**

We confirmed that all source data are correct by downloading data from the repository and recreating our datasets from the repo. Perhaps your error was related to a download error? Regardless, we tested the viability of our data posted on Dryad by downloading and displaying data.

**Manuscript mentions 2017 dataset but no folders contain 2017?**

As described above, we changed the naming conventions to be clearer, including distinguishing 2017 data.

**Global datasets for 1990, 2000, 2010, and 2015... but where are the 2000 and 2010 data?** As described above, we changed the naming conventions and explicitly provided the 2000 and 2010 data.**

**Suggestion to complement the lat/lon/date and unit info in embedded metadata?** We added this information to the data description.**

**Data description mentions change stressors and uncertainty analyses, are these included in Dryad?** Good suggestion. We added to the data repository 4 additional datasets that provide stressors.

**Manuscript comments**

**Consider adding a time index in the stressor equations for the variables that vary in stressor equations to make temporal variables more explicit.**

Thank you for this suggestion. We addressed this suggestion by including this information into Table 1 (see response to comment directly below) because we felt it was more explicit as to the time-varying stressor datasets.

**Table 1. Clarify stressors that are used for 1990-2015 and the 2017 datasets.**

Thanks -- we added year columns to Table 1 to clarify the specific years of data for each stressor and to distinguish the "change" datasets from the static dataset (representing "current" ~2017 conditions.)

**To facilitate interpretation of comparison results, It might be useful to provide an overview table or description of how datasets and/or methodology different between human footprint, temporal human pressure index, and human modification.**

This is a valuable suggestion -- we added a new table (Table 6) that describes the differences in datasets and/or methodology between our work here and other recent datasets.

**Consider changing "Hmed" to "Hmedian" to ease reading and confusion with other variables.** Thanks, done.**

**L433: Comment on the differences between the bottom line findings as compared to Ramunkutty et al. and FAO findings. Why do they differ so much?**

Good suggestion. We added a discussion of the differences with Ramunkutty et al. as detailed in our response to L476 below but elected to not cite FAO findings because those data are conducted using very different methods and scales -- though their estimates are also based on land cover and do not include intensity (further response in comment below).

**L476: Comment on the differences of your findings with HF and Hurtt et al. 2006 finding.**

Following the above comment on L433, we briefly describe the differences in our finding with HF and Ramunkutty et al. (2008). We chose to not cite Hurtt et al. (2006), as they provide summary findings largely based on Ramunkutty et al. (2008). We revised our text to read (L483-496):

"The biggest differences in rankings between the H and the HF were for temperate and broadleaf mixed forests (and see comparisons of H1k and HF in Kennedy et al. 2019a, 2019b, Riggio et al. 2020). HF was estimated to result in 12.3% modification for an earlier date (~2009; Venter et al. 2016) and is lower likely because fewer stressors were included, its additive combination method, and its strongly right-skewed distribution caused by max-value normalization. The ranks of the extent of modification by biomes, however, were very similar between H, H1k, and HF. In general, H had intermediate modification levels compared to H1k and HF: with H1k levels being slightly higher (difference between 0.00 min to 0.09 and average difference of 0.05 by biome) and HF being slightly lower (difference between 0.00 min to 0.13 max and average difference of 0.04 by biome; Table 7). Results for ecoregions shown in Fig. 1 are even more striking, as the mean annualized difference values for HF and THPI were inconsistent with our results. Of the 814 ecoregions that had increases in Hmad, a decrease in modification was found for 201 ecoregions in HFmad and 202 for THPImad; and for the 32 ecoregions that were found to have decreases in Hmad, an increase in modification was found for 20 in HF and 22 in THPI."

**Related to Reviewer #1's comment, consider for increased usability providing an "overview" a 0.5 degree resolution dataset.**

Thank you for this suggestion. While we understand these files can take long to download depending on internet access, we opted to maintain the original resolution, anticipating (from experience), that having multiple versions will lead to confusion and likely different (albeit slightly) results due to resolution differences. In this way, potential users can aggregate this dataset into multiple scales based on their specific purpose. To address the need for reduced download file size, we changed the datasets from 32-bit floating point to a 16-bit integer, which reduces each dataset by half or more (with LZW compression).

Sincerely,

David M. Theobald, Ph.D., on behalf of co-authors

**Earth transformed: detailed mapping of global human modification from 1990 to 2017**

David M. Theobald1,2, Christina Kennedy3, Bin Chen4, James Oakleaf3, Sharon Baruch-Mordo3, Joe Kiesecker3

1Conservation Planning Technologies, Fort Collins, CO 80521, USA

[revised manuscript text omitted]

(H=0.021290.0213); deserts and xeric shrublands (H=0.057060.0572); and montane grasslands and shrublands (H=0.089430.0894).

WeFollowing thresholds from Kennedy et al. (2019a), we found that in ~2017, 51.0% of global lands had very low human modification (mean  $H \le 0.01$ ; 66.8 M km2a mean value  $H \le 0.01$  (i.e. very low human modification, ), 13.3% had low human modification (a mean of  $0.01 < H \le 0.1$ ; 17.4 M km2), 21.0% had moderate human modification ( (low) , 21.0% had a mean of  $0.1 < H \le 0.4$ ; 27.6 M km2 (i.e. moderate), 12.3% had high human modification (a mean value of  $0.4 < H \le 0.7$ ; 16.1 M km2 (high), and 2.4% had very high human modification ( $0.7 < H \le 1.0$ ; 3.2 M km2) (following the thresholds from Kennedy et al. 2019a). We found that ~4.2% of lands have no evidence of human modification (H < 0.00001; 5.5 M km2), based on our estimate of the level of precision (~0.0001) given the data inputs. (3.2 M km2,  $\pm 0.0003$ ) had a mean of  $0.7 < H \le 1.0$  (very high). By area, these results by class amount to: very low=66.8 M km2 ( $\pm 0.0067$ ), low=17.4 M km2( $\pm 0.0017$ ), moderate= 27.6 M km2 ( $\pm 0.0028$ ), high=16.1 M km2 ( $\pm 0.0016$ ), and very high=3.18 M km2,  $\pm 0.0003$ ). Also, we found that 17.6% had a mean value of H < 0.0001 (23.0 M km2,  $\pm 0.0023$ ), and 4.2% had H < 0.00001 (5.5 M km2,  $\pm 0.0006$ ).

**3.3 Comparisons**

We compared our work to earlier efforts (summarized in Table 6) to determine if overall trends and extents were generally consistent and resulting with similar priorities of biomes and ecoregions were similar. Globally,  $H_{mad}$  from 1990-2015 (t=1990, u=2015) was 0.000490.0005, while for HF and THPI it was higher ( $HF_{mad}$ =0.000560.0006,  $THPI_{mad}$ =0.000810.0008). Perhaps more important is that the variability of the mean annualized difference values in the HF and THPI was 2.3 and 3.2 times that of HMH. By continent, we found that  $H_{mad}$  increased the most in Oceania, followed by Asia, Europe, Africa, South America, North America, and Australia. Continental ranks by THPI followed HMH roughly, though HF differed more substantially (Table 5).  $H_{mad}$ = increased for all continents, but  $HF_{mad}$  showed *declines* in modification for Europe and South America, while  $THPI_{mad}$  showed a decline for North America.

We also found the ranking of biomes by mean annualized difference for HF and THPI were fairly different from HM ranksranks developed from H values (Table 67). Of the three biomes with the largest increase for  $H_{mad}$ , two of them were also identified by HF (tropical & subtropical dry broadleaf forests and tropical & subtropical moist broadleaf forests) and none of them by THPI. Of the five biomes with the largest increase for  $H_{mad}$ , three of them were also identified by HF and THPI. The biomes that had the greatest disagreement amongst the ranking of HMH, HF, and THPI were mangroves; tropical & subtropical coniferous forests; and tropical & subtropical dry broadleaf forests. The results for ecoregions shown in Fig. 1 are even more striking, as the mean annualized difference values for HF and THPI were inconsistent with HM results. Of the 814 ecoregions that had increases in  $H_{mad}$ , a decrease in modification was found for 201 ecoregions in HF and 202 for THPImad; and for the 32 ecoregions that were found to have decreases in  $H_{mad}$ , an increase in modification was found for 20 in HF and 202 for THPImad; and for the 32 ecoregions that were found to have decreases in  $H_{mad}$ , an increase in modification was found for 20 in ecoregions and increase in modification was found for 20 in HF and 202 for THPImad; and for the 32 ecoregions that were found to have decreases in  $H_{mad}$ , an increase in modification was found for 20 in HF and 202 for THPImad.

**In terms of the overall amount of recent (~2017) human modification globally, we found that 14.5% of terrestrial lands globally have been modified — which is roughly similar to HF (12.3% for ~2009; Venteret al. 2016) and the degree of human modification at 1-km resolution (HM1k; 19% for ~2016; Kennedy et al. 2019a)**

The biggest differences in rankings between the H and the HF were for temperate and broadleaf mixed forests (and see comparisons of H1k and HF in Kennedy et al. 2019a, 2019b, Riggio et al. 2020). HF was estimated to result in 12.3% modification for an earlier date (~2009; Venter et al. 2016) and is lower likely because fewer stressors were included, its additive combination method, and its strongly right-skewed distribution caused by max-value normalization. The ranks of the extent of modification by biomes, however, were very similar between HM, HM1kH, H1k, and HF. In general, the HMH had intermediate modification levels compared to HM1kH1k and HF: with HM1kH1k levels being slightly higher (difference between 0.00 min to 0.09 and average difference of 0.05 by biome) and HF being slightly lower (difference between 0.00 min to 0.13 max and average difference of 0.04 by biome) (Table 6). The; Table 7). Results for ecoregions shown in Fig. 1 are even more striking, as the mean annualized difference values for HF and THPI were inconsistent with our results. Of the 814 ecoregions that had increases in  $H_{mad}$ , a decrease in modification was found for 201 ecoregions in  $HF_{mad}$  and 202 for  $THPI_{mad}$ ; and for the 32 ecoregions that were found to have decreases in  $H_{mad}$ , an increase in modification was found for 20 in HF and 22 in THPI.

Finally, the global estimate for HM1kH1k was likely higher than HM because HM1kH because H1k did not limit the livestock stressor at LU km-2 <1.0, used a slightly higher value for the low-threshold on the electrical infrastructure and energy use stressor (i.e. "nightlights"), and reported results that incorporate uncertainty in estimates of intensity. The biggest differences in rankings between the HM and the HF were for temperate and broadleaf mixed forests (and see comparisons of HM1k and HF in Kennedy et al. 2019a, 2019bFurthermore, global modification from farming was estimated at 37% for 2000 (Ramankutty et al. 2008) compared to 14.6% with H. The difference with our results is largely due to their mapping of the area land cover types but not differentiating the intensity of the impact of those cover types (crop and pasture).•

**3.4 Uncertainty and validation analyses**

To examine the uncertainty associated with our intensity estimates, we calculated across all terrestrial lands the mean *H* value on datasets generated with intensity values drawn from a uniform random distribution between the minimum and maximum estimates. We generated 50 randomized datasets and found the mean of the randomized maps was 0.14306 and the standard deviation was  $\pm 0.00106$  (compared to We addressed uncertainty in our results by incorporating the parameter  $p(C_s)$  for every sector *s* to best quantify uncertainty in its spatial location and classification as detailed in section 2.2.; for example, we adjusted  $p(C_{cp})$  by directly incorporating measured confusion among land cover types using the results from the accuracy assessment of the land cover dataset (from Eq. 4). Additionally, we incorporated uncertainty by calculating the global mean for each of the 50 randomizations, which across the 50 iterations was 0.1434 (SD=  $\pm 0.0076$ ) and ranged from 0.1243 to 0.1612. Thus, the global mean of 0.1461 obtained using our "best-estimate of 0.14605). The lowestpossible mean *H* value calculated with the minimum estimate for all stressors was 0.10686 and the highest possible value using the maximum estimate was 0.18493. " intensity values was in line with our uncertainty results. We also mapped the per-pixel variance (standard deviation) to examine the spatial pattern of uncertainty (Figure 4). The locations of the highest levels of uncertainty tend to be in more highly developed landscapes.

[revised manuscript text omitted]

**4.2 Caveats**

As with any model, we recognize there are limitations of our work. We did not include data for all human stressors, typicallylargely because of incomplete global coverage or too-coarse mapping units (Klein Goldewijk et al. 2007; Geldmann *et al.*, 2014), an inability to discern human-induced versus natural disturbances (e.g.; wildfires), or uncertainty in the location and directionality of its impact (e.g.; climate change on terrestrial systems; Geldmann *et al.*, 2014). Although our datasets described-here have order of magnitude higher resolution than previous temporal maps, estimates of *H* generated for areas less than roughly 100 km2 should be used with caution. In particular and discussed in Kennedy et al. (2019a, 2019b), changes to land cover due to ecological disturbance events, such as wildfires or flooding, are not included in our analysis because of the difficulty in separating natural from human-caused disturbances – yet, we recognize that the broad extent of wildfire in particular, could have strong implications. We did not include climate data as a stressor in this product to keep our analysis manageable and tractable. For more integrated analyses, our data product should be used in combination with datasets of impacts due to climate change (e.g., Parks et al. 2020).

Stressors that are particularly important to improve include effects of grazing (currently coarse data and very broad expanse), pasture land, invasive species, and climate change (especially wildfire and effects of sea-level rise), and we encourage future work to focus on developing appropriate datasets and approaches to include or better capture these stressors. Key datasets we believe should be improved include transportation networks•, including logging roads (e.g., Van Etten 2019) that are comparable through time; livestock grazing,= rangelands, croplands, timber plantations, and pasturelands and their intensity of use; resource extraction (especially mining footprints); and temporal trends in gas flares, utility-scale solar plants, electrical substations, etc.¶

**4.3 Data availability**

The datasets generated from this work are available here (Figshare DOI pending). All other datasets used in our work are open source data and are listed below.¶

f

- Center for International Earth Science Information Network: Global Rural-Urban Mapping Project, Version 1 (GRUMPv1): Settlement Points, Revision 01. Palisades, NY: NASA Socioeconomic-Data and Applications Center (SEDAC). https://doi.org/10.7927/H4BC3WG1, 2017, Accessed 1 January 2019.¶
- Center for International Earth Science Information Network: Gridded Population of the World, Version 4 (GPWv4): Administrative Unit Center Points with Population Estimates, Revision 11. Palisades, NY: NASA Socioeconomic Data and Applications Center (SEDAC). https://doi.org/10.7927/II4BC3WMT, 2018, Accessed 1 January 2019.¶
- Corbane, C., Florczyk, A., Pesaresi, M., Politis, P., Syrris, V.: GHS built-up grid, derived from Landsat, multitemporal (1975-1990-2000-2014), R2018A. European Commission, Joint Research Centre-(JRC) doi: 10.2905/jrc-ghsl-10007 PID: http://data.europa.eu/89h/jrc-ghsl-10007, 2018¶
- Crippa, M., Guizzardi, D., Muntean, M., Schaaf, E., Dentener, F., van Aardenne, J. A., Janssens-Maenhout, G.: Gridded emissions of air pollutants for the period 1970–2012 within EDGAR v4. 3.2. Earth System Science Data, 10(4), 1987-2013, https://data.jrc.ec.europa.eu/collection/EDGAR, 2018.¶
- Curtis, P.G., Slay, C.M., Harris, N.L., Tyukavina, A., Hansen, M.C.: Classifying drivers of global forestloss. Science, 361(6407), pp.1108-1111,
  - https://science.sciencemag.org/content/sci/suppl/2018/09/12/361.6407.1108.DC1, 2018.¶
- Elvidge, C., Ziskin, D., Baugh, K., Tuttle, B., Ghosh, T., Pack, D., Erwin, E., Zhizhin, M.: A Fifteen Year Record of Global Natural Gas Flaring Derived from Satellite Data. Energies 2 (3): 595, doi:10.3390/en20300595. https://payneinstitute.mines.edu/eog/dmsp/, 2009.
- Geldmann, Jonas; Joppa, Lucas; Burgess, Neil D.: Temporal Human Pressure Index, v2, Dryad, Dataset, https://doi.org/10.5061/dryad.p8cz8w9kf, 2019b.¶
- Gilbert, M., Nicolas, G., Cinardi, G., Van Boeckel, T. P., Vanwambeke, S. O., Wint, G. W., Robinson, T. P. Global distribution data for cattle, buffaloes, horses, sheep, goats, pigs, chickens and ducks in 2010. Scientific Data, 5, 180227, <a href="https://doi.org/10.7910/DVN/BLWPZN">https://doi.org/10.7910/DVN/BLWPZN</a>, <a href="https://doi.org/10.7910/DVN/33N0JG">https://doi.org/10.7910/DVN/BLWPZN</a>, <a href="https://doi.org/10.7910/DVN/33N0JG">https://doi.org/10.7910/DVN/BLWPZN</a>, <a href="https://doi.org/10.7910/DVN/33N0JG">https://doi.org/10.7910/DVN/33N0JG</a>, <a href="https://doi.org/10.7910/DVN/7Q52MV">https://doi.org/10.7910/DVN/7Q52MV</a>, <a href="https://doi.org/10.7910/DVN/33N0JG">https://doi.org/10.7910/DVN/7Q52MV</a>, <a href="https://doi.org/10.7910/DVN/33N0JG">https://doi.org/10.7910/DVN/7Q52MV</a>, <a href="https://doi.org/10.7910/DVN/33N0JG">https://doi.org/10.7910/DVN/7Q52MV</a>, <a href="https://doi.org/10.7910/DVN/33N0JG">https://doi.org/10.7910/DVN/7Q52MV</a>, <a href="https://doi.org/10.7910/DVN/33N0JG">https://doi.org/10.7910/DVN/7Q52MV</a>, <a href="https://doi.org/10.7910/DVN/GIVQ75">https://doi.org/10.7910/DVN/33N0JG</a>, <a href="https://doi.org/10.7910/DVN/SUFASE">https://doi.org/10.7910/DVN/SUFASE</a>, <a href="
  - https://doi:10.7910/DVN/5U8MWI, 2018.

[revised manuscript text omitted]

1Conservation Planning Technologies, Fort Collins, CO 80521, USA

2Department of Fish, Wildlife, and Conservation Biology, Colorado State University, Fort Collins, CO 80523, USA

3Global Lands Program, The Nature Conservancy, Fort Collins, CO 80524, USA

4Department of Land, Air and Water Resources, University of California, Davis, CA 95616, USA

**Tables**

**¶**

Table 1. Overview of stressors, datasets, spatial resolution, and years data were available and used in the Human Modification maps maps of human modification. Stressor classification levels in parentheses correspond to those within the Direct Threats Classification v2 (Salafsky et al. 2008).

| <del>Class¶</del>                                  | Stressor*¶                                                           | Source¶                                                                                                                                                         | Resolution
(km²)¶                                     | <del>Year(s)¶</del>                                                          |
|----------------------------------------------------|----------------------------------------------------------------------|-----------------------------------------------------------------------------------------------------------------------------------------------------------------|----------------------------------------------------------|------------------------------------------------------------------------------|
| <del>Urban &</del>
<del>built-up (1)¶</del> | <del>-Built-up (1.1,</del>
<del>1.2)¶</del>                       | Global Human Settlement Layer-
version R2018A (Pesaresi et al. 2015)¶                                                                                        | <del>0.0009-0.9</del>                                    | <del>1990, 2000,</del>
<del>2010*, 2015¶</del>                 |
| <del>Agriculture</del>
<del>(≥)¶</del>          | Croplands &
pasturelands
(2.1)                                 | European Space Agency CCI land
cover (Li et al. 2018)¶
Unified Cropland Layer (Waldner et
al. 2016)¶
Global Land Systems v2 (Kehoe et
al. 2017)¶ | <del>0.9¶</del>
¶
<del>1¶</del>
¶
1 ¶ | <del>1992, 2000,</del>
<del>2010, 2015¶</del>
2010¶
¶
2010¶
¶ |
|                                                    | <del>Grazing (2.3)¶</del>                                            | Gridded Livestock v3 (Robinson et
al. 2014; Gilbert et al. 2018)¶                                                                                            | <del>10¶</del>                                           | <del>2010¶</del>                                                             |
| Energy-
production
& mining (3)¶             | Oil & gas-
production (3.1)¶                                      | Nighttime flares from DMSP/OLS-
and VIIRS (Elvidge et al. 2009;
Elvidge et al. 2016)¶
¶                                                                | <del>0.25 - 1.0¶</del>                                   | <del>2016¶</del>                                                             |
|                                                    | Mining &
quarrying (3.2)¶                                         | <del>S&P global mining dataset (S&P</del>
<del>2018; Valenta et al. 2019)¶</del>                                                                     | <del>~1:10000¶</del>                                     | <del>1990, 2000,</del>
<del>2010, 2015,</del>
<del>2018¶</del>         |
|                                                    | Renewable (3.3)
and
non-renewable
power (1.2)
generation | World Resources Institute Power-
plants (WRI 2019)¶                                                                                                          | <del>~1:100000¶</del>                                    | <del>1990, 2000,</del>
<del>2010, 2015,</del>
<del>2018¶</del>         |
| Transportati
on & service                       | Roads (4.1)¶                                                         | OSM highway, minor, and two-track
features (OpenStreetMap 2019)¶                                                                                             | <del>~1:10-25000¶</del>                                  | <mark>2019¶</mark>                                                           |
| corridors (4)                                      | Railways (4.1)¶                                                      | <del>OSM railway features-
(OpenStreetMap 2019)¶</del>                                                                                                      | <del>~1:10-25000¶</del>                                  | <del>2019¶</del>                                                             |
|                                                    | Powerlines (4.2)                                                     | <del>OSM power line features</del>
<del>(OpenStreetMap 2019)</del> ¶                                                                                         | 1:10-25000¶                                              | <del>2019¶</del>                                                             |

| f                                             | Electrical-
infrastructure-
(4.2)¶      | Nighttime lights from DMSP/OLS
and VIIRS (Elvidge et al. 2001; Doll-
2008; Elvidge et al. 2013; Zhang et-
al. 2016)¶         | <del>0.25-1.0¶</del> | 1992, 2000,
2010, 2015,
2018¶               |
|-----------------------------------------------|-----------------------------------------------|---------------------------------------------------------------------------------------------------------------------------------------|----------------------|---------------------------------------------------|
| Biological
harvesting
(5)¶              | Logging &
wood
harvesting (5.3)¶        | Forest loss (Curtis et al. 2018) and
forest change (Hansen et al. 2013)¶                                                           | <del>0.09−100¶</del> | <del>2000, 2010,</del>
<del>2018¶</del>        |
| Human-
intrusions-
(6)¶                 | Human-
intrusions (1.3,
5.1, 5.2, 6.1)¶ | Human intrusion (Theobald 2008)-
using accessibility and population
from GRUMP v1.01 (CIESIN 2017)
and GPW v4 (CIESIN 2018)¶ | 11                   | 1990*,
2000, 2010,
2015¶                    |
| Natural-
system-
modification
s-(7)¶ | <del>Reservoirs (7.2)¶</del>
¶             | Global Reservoirs and Dams (GRanD-
v1.3; Lehner et al. 2011);
http://globaldamwatch.org/grand/¶                                 | <del>1:25000¶</del>  | <del>1990, 2000,</del>
<del>2010, 2017¶</del>  |
| Pollution (9)¶                                | Air pollution
(9-5)¶                       | Emissions Database for Global
Atmospheric Research (EDGAR
v4.3.2; Crippa et al. 2018) for
nitrogen oxides                    | ~ <del>100</del> ¶   | <del>1990, 2000,</del>
<del>2010, 2012</del> ¶ |

Acronyms of source data are bolded in the Source column for reference throughout the paper. For each stressor, the years 1990-2015 are used for change analysis, and ~2017 is a compilation of all stressors that represents "current" conditions with the median year of 2017.

| Class                   | Class Stressor* Source Scale   |                                                                                                                                                                                                                                        | Scale           |      |      | Year                 |      |                      |
|-------------------------|--------------------------------|----------------------------------------------------------------------------------------------------------------------------------------------------------------------------------------------------------------------------------------|-----------------|------|------|----------------------|------|----------------------|
|                         |                                |                                                                                                                                                                                                                                        | (km²)           | 1990 | 2000 | 2010                 | 2015 | ~2017                |
| Urban &
built-up (1) | Built-up (1.1,
1.2)         | Global Human Settlement
Layer version R2018A
( GHSL; Pesaresi et al. 2015)                                                                                                                                                | 0.0009
- 0.9 | 1990 | 2000 | 2010 *    | 2015 | 2015                 |
| Agriculture
(2)      | Croplands & pasturelands (2.1) | European Space Agency
Climate Change Initiative
land cover ( ESA CCI ; Li et
al. 2018)
Unified Cropland Layer
( UCL ; Waldner et al. 2016)
Global Land Systems v2
( GLS ; Kehoe et al. 2017) | 0.9
1
1   | 1992 | 2000 | 2010
2010
2010 | 2015 | 2015
2010
2010 |
|                         | Grazing (2.3)                  | Gridded Livestock of the
World v3 ( GLW ; Robinson
et al. 2014; Gilbert et al.
2018a, Gilbert et al. 2018b)                                                                                                            | 10              |      |      | 2010                 |      | 2010                 |

| Energy
production
& mining
(3)      | Oil & gas
production
(3.1)                                                                                                   | Nighttime flares from
Defense Meteorological
Program/Operational
Line-scan System
( DMSP/OLS , Elvidge et al.
2009) and Visible Infrared
Imaging Radiometer Suite
( VIIRS , Elvidge et al. 2016) |                 |       |      |      |      | 2016 |
|----------------------------------------------|------------------------------------------------------------------------------------------------------------------------------------|-------------------------------------------------------------------------------------------------------------------------------------------------------------------------------------------------------------------------------------|-----------------|-------|------|------|------|------|
|                                              | Mining &
quarrying
(3.2)                                                                                                     | S&P global mining dataset
(S&P 2018; Valenta et al.
2019)                                                                                                                                                                     | ~1:100
00    | 1990, | 2000 | 2010 | 2015 | 2018 |
|                                              | Renewable
(3.3) &
non-renewab
le power (1.2)
generation                                                                | World Resources Institute
Power plants ( WRI ; WRI
2019)                                                                                                                                                               | ~1:100
000   | 1990  | 2000 | 2010 | 2015 | 2018 |
| Transporta
tion &
service
corridors | Roads (4.1)                                                                                                                        | OpenStreetMap highway,
minor, and two-track
features ( OSM ;
OpenStreetMap 2019)                                                                                                                                    | ~1:10-2
5000 | **    |      |      | **   | 2019 |
| (4)                                          | Railways
(4.1)                                                                                                                  | OSM railway features
(OpenStreetMap 2019)                                                                                                                                                                                        | ~1:10-2
5000 |       |      |      |      | 2019 |
|                                              | Powerlines
(4.2)                                                                                                                | OSM power line features
(OpenStreetMap 2019)                                                                                                                                                                                     | ~1:10-2
5000 |       |      |      |      | 2019 |
|                                              | Electrical
infrastructur
e (4.2)                                                                                             | Nighttime lights from
DMSP/OLS and VIIRS
(Elvidge et al. 2001; Doll
2008; Elvidge et al. 2013;
Zhang et al. 2016)                                                                                                       | 0.25 -
1.0   | 1992  | 2000 | 2010 | 2015 | 2018 |
| Biological
harvesting
(5)              | Biological
harvesting
(5)Logging &
woodForest loss (Curtis et al.
2018) and forest change
(Hansen et al. 2013)(5.3) |                                                                                                                                                                                                                                     | 0.09 -
100   | 2000  | 2000 | 2010 | 2015 | 2018 |
| Human
intrusions
(6)                   | Human
intrusions
(1.3, 5.1, 5.2,
6.1)                                                                                     | Human intrusion
(Theobald 2008, HUE )
using accessibility and
population from Global
Rural-Urban Mapping
Project v1.01 ( GRUMP ;
CIESIN 2017) and Gridded                                           | 1               | 1990* | 2000 | 2010 | 2015 | 2015 |

|                                            |                        | Population of the World v4
( GPW; CIESIN 2018)                                                                   |              |      |      |      |      |      |
|--------------------------------------------|------------------------|----------------------------------------------------------------------------------------------------------------------------|--------------|------|------|------|------|------|
| Natural
system
modificatio
ns (7) | Reservoirs
(7.2)    | Global Reservoirs and
Dams ( GRanD v1.3; Lehner
et al. 2011)                                                  | ~1:250
00 | 1990 | 2000 | 2010 | 2015 | 2017 |
| Pollution
(9)                           | Air pollution
(9.5) | Emissions Database for
Global Atmospheric
Research ( EDGAR v4.3.2;
Crippa et al. 2018) for NO x | ~100         | 1990 | 2000 | 2010 | 2012 | 2012 |

\*Based on interpolation.

\*\*Used major roads (i.e. highways) for 2019.

Table 2. Estimates of the intensity value for each stressor. "Best" estimates were determined from Brown and Vivas (2005)1, Theobald (2013)2, Kennedy et al. (2019a)3, or expert judgement4, and are bracketed by a minimum and maximum range, following the lowest-highest-best estimate elicitation procedure to reduce bias (McBride *et al.*, 2012). Results presented here use the best estimate, while minimum and maximum estimates are used to specify the range of possible randomized intensity values in the uncertainty analysis.

| Class                               | Stressor                                                                                                                                               | Minimum              | Best                 | Maximum              |
|-------------------------------------|--------------------------------------------------------------------------------------------------------------------------------------------------------|----------------------|----------------------|----------------------|
| Urban & built-up                    | Built-up areas 3,4                                                                                                                          | 0.69                 | 0.85                 | 1.00                 |
| Agriculture                         | Cropland/pasture 3
- Minimal 4
- Light 4
- Intense 1,4                                            | 0.29
0.35
0.60 | 0.34
0.45
0.65 | 0.39
0.55
0.70 |
|                                     | Livestock grazing 1                                                                                                                         | 0.20                 | 0.28                 | 0.37                 |
| Energy production                   | Oil & gas production 1,3                                                                                                                    | 0.70                 | 0.85                 | 1.00                 |
| a mining                            | Mining 3                                                                                                                                    | 0.83                 | 0.91                 | 1.00                 |
|                                     | Power generation 1
(non-renewable)                                                                                                       | 0.70                 | 0.85                 | 1.00                 |
|                                     | Power generation (renewable) 1                                                                                                              | 0.70                 | 0.80                 | 0.90                 |
| Transportation & service corridors* | Major roads 1                                                                                                                               | 0.78
(20)         | 0.80
(30)         | 0.83
(40)         |
|                                     | Minor roads 1                                                                                                                               | 0.39
(15)         | 0.44
(20)         | 0.50
(25)         |
|                                     | Two-track roads 1                                                                                                                           | 0.10
(3)          | 0.15
(5)          | 0.20
(10)         |
|                                     | Railways 1                                                                                                                                  | 0.78
(15)         | 0.80
(20)         | 0.83
(25)         |
|                                     | Powerlines 2                                                                                                                                | 0.10                 | 0.15                 | 0.20                 |
|                                     | Electrical infrastructure (night-time lights) 3                                                                                             | 0.20                 | 0.35                 | 0.50                 |
| Biological
harvesting            | Logging & wood harvesting 1,4 **
- Commodity-driven 1,4
- Shifting agriculture 1,4
- Forestry 1,4 | 0.60
0.10
0.10 | 0.65
0.20
0.20 | 0.07
0.30
0.30 |

[revised manuscript text omitted]

¶

Table 6. A summary of the data, methods, and results comparing the degree of human modification (HM; this paper); degree of human modification 1 km (HM1k; Kennedy et al. 2019a, 2019b, 2019c); human footprint (HF; Sanderson et al. 2002; Venter et al. 2016); and temporal human pressure index (THPI; Geldmann et al. 2019). Also see discussion of comparison in Kennedy et al. (2019b, 2019c), Venter et al. (2019), and Riggio et al. (2020). Data source acronyms are provided in Table 1.

| Factor                                                 | НМ                                                                                                                                                                                                              | HM1k                                                                                                                                                                                               | HF                                                                                                               | ТНРІ                                                             |
|--------------------------------------------------------|-----------------------------------------------------------------------------------------------------------------------------------------------------------------------------------------------------------------|----------------------------------------------------------------------------------------------------------------------------------------------------------------------------------------------------|------------------------------------------------------------------------------------------------------------------|------------------------------------------------------------------|
| Conceptual
framework                                | Direct Threats Classification v2
(Salafsky et al. 2008)
Intensity values based on Land
Development Index (LDI; Brown
and Vivas 2005)                                                                | Direct Threats Classification
v2 (Salafsky et al. 2008)
LDI                                                                                                                                  | Sanderson et al.
(2002)                                                                                       | Geldmann et al.
(2014)                                        |
| Stressor:
Urban and
built-up                     | Urban and built-up (GHSL;
0.03-0.3 km; 1990-2015)                                                                                                                                                            | Urban and built-up (GHSL;
0.03-0.3 km; 2015)
Population density (GPW v4
2015, 1 km)                                                                                                       | Night-time lights
(DMSP/OLS >20; 1 km;
1994-2012)
Population density
CIESIN v3; 4 km; 1990,
2010) | Change in
population density
(GPW v3 1995,
2010, 1 km)  |
| Stressor:
Agriculture                               | Cropland & pastureland for 1990,
2015 (ESA CCI; 300 m) and
Cropland intensity (GLS, 1 km)
Unified Cropland Layer (UCL, 1
km)
Grazing (GLW, 10 km, 1 < livestock
units/km 2 < 1000) | Unified Cropland Layer
(UCL, 1 km)
Grazing (GLW v2, 1 km,
livestock units/km² < 1000)                                                                                                     | Cropland (UMD for
1990 and GlobCover
for 2009);
Pastureland (2000),
10 km                            | Cropland area
(HYDE, 10 km)                                   |
| Stressor:
Energy
production &
mining          | Oil & gas production (Gas flares
DMSP/OLS and VIIRS)
Renewable and non-renewable
power plants (WRI)
Large mining operations (S&P)                                                                   | Oil & gas wells, wind
turbines, mines (OSM, 2016,
VMAP0-2000)                                                                                                                                | N/A                                                                                                              | N/A                                                              |
| Stressor:
Transportatio
n & service
corridors | Road (highway, minor, two-track;
OSM, 2019)
Railways (OSM, 2019) Powerlines
(OSM, 2019)
Electrical power infrastructure
(harmonized DMSP/ VIIRS,
1992-2018)                                   | Road (highway, minor,
two-track, OSM, 2016,
gROADS-2000)
Railways (OSM, 2016,
VMAP0-2000), Powerlines
(OSM 2016)
Electric infrastructure
(night-time lights
DMSP-OLS-2013) | Roads (gROADS,
1980-2000); Railways
(VMAP-2000)
Electric infrastructure                                 | Nightlights
(DMSP/OLS
nightlights >20; 1
km; 1994-2012) |
| Stressor:
Biological
harvesting                  | Forest loss (Hansen, Curtis; 0.03-1
km, 2000-2017)                                                                                                                                                           | N/A                                                                                                                                                                                                | N/A                                                                                                              | N/A                                                              |
| Stressor:
Human
intrusions                       | Human intrusion (HUE, 1990-2015,
1 km)                                                                                                                                                                       | N/A                                                                                                                                                                                                | N/A                                                                                                              | N/A                                                              |
| Stressor:                                              | Reservoirs (GRanD, 1990-2017,                                                                                                                                                                                   | N/A                                                                                                                                                                                                | N/A                                                                                                              | N/A                                                              |

| Natural
system
modifications        | 0.03 km)                                                                                                             |                                                                                                                                     |                                                                                                 |                                         |
|-------------------------------------------|----------------------------------------------------------------------------------------------------------------------|-------------------------------------------------------------------------------------------------------------------------------------|-------------------------------------------------------------------------------------------------|-----------------------------------------|
| Stressor:
Pollution                    | Nitrous oxide pollution (EDGAR,
1990-2012, 100 km)                                                                | N/A                                                                                                                                 | N/A                                                                                             | N/A                                     |
| Metric                                    | Degree of human modification (H,
0-1.0 continuous value)                                                          | Degree of human
modification (H, 0-1.0
continuous value)                                                                      | Scaled 0-10, 0-4,
summed to 50,
ordinal value                                             | N/A                                     |
| Combine
factors                        | Increasive to 1.0 using fuzzy sum                                                                                    | Increasive to 1.0 using fuzzy sum                                                                                                   | Additive,
max-normalized                                                                     | Equal-weight,
additive
normalized |
| Uncertainty
or sensitivity
analysis | Calculates per-pixel variance due
to estimates of intensity values,
randomized (n=50)                          | Calculates per-pixel variance
due to estimates of
intensity values,
randomized (n=100)                                     | Sensitify of static v.
dynamic pasture data                                                  | N/A                                     |
| Validation                                | Tested using independent
validation dataset that included
~10,000 subplots within ~1,000 1
km² sample plots | Tested using independent
validation dataset that
included ~10,000 subplots
within ~1,000 1 km 2 sample
plots | Tested using
independent
validation dataset in
3,460 1 km 2 sample
plots | N/A                                     |

Table 67. Summary of results by biome, comparing trends using the mean annualized difference for the human modification ( $HM_{mad}H_{mad}$ ), human footprint ( $HF_{mad}$ , Venter et al. 2016), and the mean temporal human pressure index ( $THPI_{mad}$ , Geldmann et al. 2019) score. Also provided are estimates of the proportion of terrestrial lands modified as estimated by HM, human modification from Kennedy et al. (HM1kH1k; 2019), and HF (score was max-normalized to rescale to 0-1). The THPI dataset only-characterizes characterizes only change and so estimates of the proportion of lands modified in 2010 could not be provided. Mean annualized mean difference is calculated as the mean value across the continents and globally of the difference in H values divided by the number of years.

| Biome name                           | HM mad HM
(1990-2015) | HF mad HF
(1993-2009) | THPI mad TH
PI*
(1995-2010) | HM
(~2017) | HM1kH
M**
(~2016) | HF
(2009) |
|--------------------------------------|-------------------------------------|-------------------------------------|----------------------------------------------|---------------|-------------------------|--------------|
|                                      | 0.000004                            | - <del>0.000014</del>               | 0.000001                                     |               |                         |              |
| Boreal Forests/Taiga                 | 0.00000                             | 0.00001                             | 0.00000                                      | 0.0213        | 0.0374                  | 0.0288       |
|                                      | 0.0000100                           | 0.000028                            | 0.000032                                     |               |                         |              |
| Deserts & Xeric Shrublands           | .00001                              | 0.00003                             | 0.00003                                      | 0.0571        | 0.1059                  | 0.0820       |
|                                      | 0.0000220                           | <del>0.000023</del> 0               | <del>0.0001520</del>                         |               |                         |              |
| Flooded Grasslands & Savannas        | .00002                              | .00002                              | .00015                                       | 0.2024        | 0.2480                  | 0.1423       |
|                                      | 0.000050                            | <del>0.000047</del>                 | <del>0.000021</del> 0                        |               |                         |              |
| Mangroves                            | 0.00005                             | 0.00005                             | .00002                                       | 0.2165        | 0.3051                  | 0.1972       |
| Mediterranean Forests, Woodlands &   | 0.000033                            | <del>0.000078</del>                 | 0.0001200                                    |               |                         |              |
| Scrub                                | 0.00003                             | 0.00008                             | .00012                                       | 0.2903        | 0.3373                  | 0.2162       |
|                                      | <del>0.000013</del> 0               | 0.000059                            | 0.000057                                     |               |                         |              |
| Montane Grasslands & Shrublands      | .00001                              | 0.00006                             | 0.00006                                      | 0.0894        | 0.1634                  | 0.1076       |
|                                      | <del>0.000023</del> 0               | <del>0.000027</del> 0               | 0.000022                                     |               |                         |              |
| Temperate Broadleaf & Mixed Forests  | .00002                              | .00003                              | 0.00002                                      | 0.3744        | 0.3968                  | 0.2485       |
|                                      | <del>0.0000160</del>                | 0.0000110                           | 0.000057                                     |               |                         |              |
| Temperate Conifer Forests            | .00002                              | .00001                              | 0.00006                                      | 0.1072        | 0.1561                  | 0.0992       |
| Temperate Grasslands, Savannas &     | <del>0.000015</del> 0               | 0.000006                            | <del>0.000092</del>                          |               |                         |              |
| Shrublands                           | .00002                              | 0.00001                             | 0.00009                                      | 0.2374        | 0.2943                  | 0.1668       |
| Tropical & Subtropical Coniferous    | <del>0.000032</del> 0               | 0.000005                            | 0.000247                                     |               |                         |              |
| Forests                              | .00003                              | 0.00000                             | 0.00025                                      | 0.2052        | 0.2606                  | 0.1568       |
| Tropical & Subtropical Dry Broadleaf | 0.000046                            | <del>0.000118</del> 0               | <del>0.000056</del>                          |               |                         |              |
| Forests                              | 0.00005                             | .00012                              | 0.00006                                      | 0.3317        | 0.4242                  | 0.2265       |
| Tropical & Subtropical Grasslands,   | 0.000020                            | 0.000057                            | <del>0.000084</del>                          |               |                         |              |
| Savannas & Shrublands                | 0.00002                             | 0.00006                             | 0.00008                                      | 0.1476        | 0.2120                  | 0.1207       |
| Tropical & Subtropical Moist         | 0.000047                            | <del>0.000074</del>                 | 0.000092                                     |               |                         |              |
| Broadleaf Forests                    | 0.00005                             | 0.00007                             | 0.00009                                      | 0.1862        | 0.2310                  | 0.1390       |
| Tundra                               | 0.0000010                           | <del>0.0000010</del>                | -0.000001                                    | 0.0023        | 0.0001                  | 0.0066       |

|   | .00000 | .00000 | 0.00000 |  |  |
|---|--------|--------|---------|--|--|
| f |        |        |         |  |  |
| f |        |        |         |  |  |
| ¶ |        |        |         |  |  |
| ¶ |        |        |         |  |  |

**Figure captions**

Figure 1. A comparison of the recent trends in human activities by ecoregion using the mean annualized difference estimated by: (a) human modification (*H*, from 1990-2015); (b) human footprint (for 1993-2009, Venter et al. 2016); and (c) temporal human pressure index (for ~1995-2010, Geldmann et al. 2019). Note: interactive maps are available hereat: https://davidtheobald8.users.earthengine.app/view/global-human-modification-change.

---

## Author Response (AR2)

July 28, 2020

Dear Editor David Carlson,

Thank you for your handling of our manuscript ESSD-2019-252. In response to the short comment from Reviewer #2, we added a sentence in the Caveat section:
"Although we attempted to map each stressor comprehensively, we recognize that some datasets may have missing features, particularly for mine and oil/gas wells -- though large mines and concentrated oil/gas fields have been mapped quite well. "

Also, we updated the Zenodo repository and so updated in the manuscript the citation and links to the repository.

Many thanks!

Sincerely,

David M. Theobald, Ph.D., on behalf of co-authors